# Plasmon-driven chemical transformation of a secondary amide probed by surface enhanced Raman scattering
Anushree Dutta[1,2], Milan Ončák[3], Farhad Izadi[3], Eugene Arthur-Baidoo[3], João Ameixa[1], Stephan Denifl[3] & Ilko Bald[1,4] ✉

Plasmon-driven chemical conversion is gaining burgeoning interest in the field of heterogeneous catalysis. Herein, we study the reactivity of N-methyl-4-sulfanylbenzamide (NMSB) at nanocavities of gold and silver nanoparticle aggregates under plasmonic excitation to gain understanding of the respective reaction mechanism. NMSB is a secondary amide, which is a frequent binding motive found in peptides and a common coupling product of organic molecules and biomolecules. Surface-enhanced Raman scattering (SERS) is used as a two-in-one in-situ spectroscopic tool to initiate the molecular transformation process and simultaneously monitor and analyze the reaction products. Supported by dissociative electron attachment (DEA) studies with the gas phase molecule, a hot electron-mediated conversion of NMSB to p-mercaptobenzamide and p-mercaptobenzonitrile is proposed at the plasmonic nanocavities. The reaction rate showed negligible dependence on the external temperature, ruling out the dominant role of heat in the chemical transformation at the plasmonic interface. This is reflected in the absence of a superlinear relationship between the reaction rate constant and the laser power density, and DEA and SERS studies indicate a hot-electron mediated pathway. We conclude that the overall reaction rate is limited by the availability of energetic hot electrons to the NMSB molecule.

Photocatalysis at plasmonic interfaces driven by surface plasmons has emerged as a promising approach to facilitate light-driven chemical conversions of molecules under mild conditions at longer wavelengths than conventional photochemistry[1,2]. Surface plasmons are collective oscillations of free electrons at the interface of a metal and a dielectric. Importantly, the decay of the surface plasmons through electron-electron scattering results in the generation of energetic electrons above the Fermi level called "hot-electrons"[3,4]. These potentially energetic electrons can readily facilitate chemical reactions on nanoparticle surfaces opening up the field of plasmon chemistry. The ability of plasmon chemistry to study the reactivity of small molecules ($NH_3$, $H_2$, $O_2$, etc.) and model organic molecules, viz., nitroarenes, aryl thiols, aryl halides, etc. have been well-proven in several studies[5–13]. This has been extended further to study the reactivity of a range of biomolecules like peptides in a plasmonic hot spot by tip-enhanced Raman scattering (TERS)[14].

A common explanation of plasmon-induced processes is based on the well-developed models for reactions induced by electronic transitions namely desorption induced by electronic transition (DIET), and desorption induced by multiple electronic transitions (DIMET). DIET involves the formation of transient negative ions (TNIs), formed upon electron transfer from metal to adsorbates[6,14,15]. While the DIET and DIMET mechanisms assume a reversible electron transfer resulting in vibrationally excited neutral molecules, the TNIs can also undergo nonergodic bond dissociation along a repulsive potential, following a non-thermal pathway to result in the product species. A similar mechanism for plasmon-driven photochemistry which follows the dissociative electron attachment (DEA) mechanism has been proposed in our previous studies[12,16]. Importantly, the probability of electron transition-driven on-surface chemical transformations is guided by many factors including the availability of low-lying molecular orbitals of the adsorbate on the metal, excitation wavelength, power density of the incident light, the potential energy landscape of the adsorbate, the surface temperature of the plasmonic material, interfacial chemical factors, etc[3,6,17–21].

As proposed by Linic and coworkers, injection of hot carriers (hot electrons) into molecules can occur via two different mechanisms – direct

[1]Institute of Chemistry, University of Potsdam, Karl-Liebknecht-Str. 24-25, 14476 Potsdam, Germany. [2]Department of Chemistry and Applied Biosciences, ETH Zurich, Zurich, CH-8093, Switzerland. [3]Institut für Ionenphysik und Angewandte Physik, Universität Innsbruck, Technikerstraße 25, 6020 Innsbruck, Austria. [4]J. Heyrovský Institute of Physical Chemistry of the CAS, Dolejškova 3, Prague, 18223, Czech Republic. ✉e-mail: bald@uni-potsdam.de

**Fig. 1 | SEM images of Ag and Au aggregates and cartoon depictions of the chemical transformations occurring at the plasmonic nanocavities of the aggregates. A, B** SEM images of Ag and **C** that of Au aggregates coated with NMSB molecule deposited on Si-wafer; **D** cartoon depicting the honeycomb arrangement of the Ag and Au aggregates with NMSB molecules, I (panel **D**1 – enlarged view) and the subsequent products, II - MBAm and III – MBN (panel **D**2 – enlarged view) formed in the plasmonic nanocavities upon visible light (633 nm and 785 nm) irradiation.

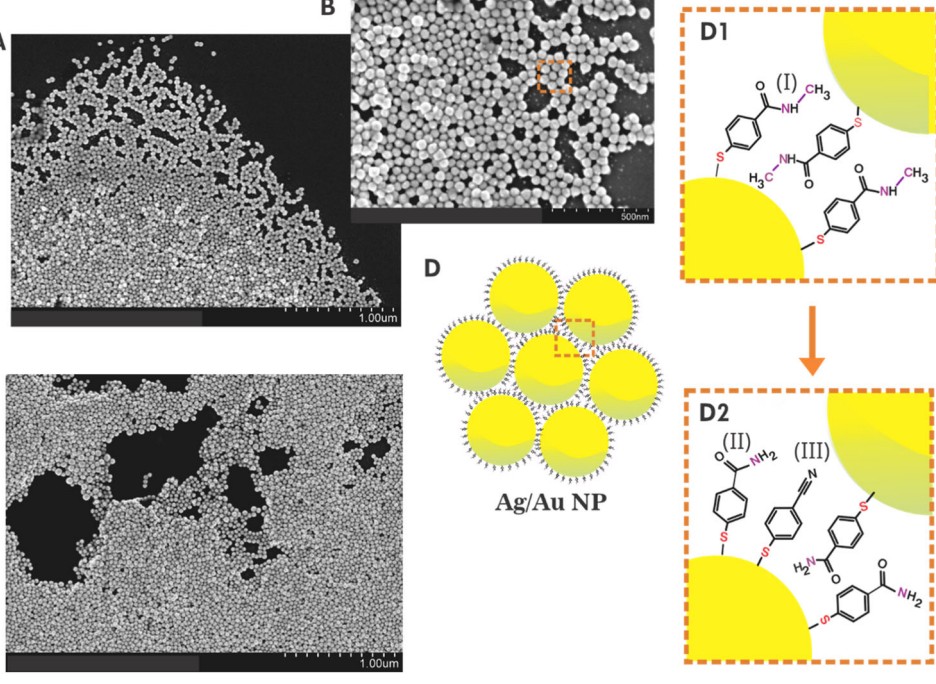

(chemical interface damping) and indirect (Landau damping) charge transfer[22]. Therefore, the success of charge transfer-induced molecular transformation depends highly on the accessibility of molecular orbitals of the adsorbate. That is to say, that the frontier orbitals, highest occupied molecular orbitals (HOMO), and lowest unoccupied molecular orbitals (LUMO) lying in the accessible range around the Fermi level, can participate in the surface reaction[1]. In summary, light absorption by the metal might result in energy transfer to the adsorbate – either via phonon coupling (resulting in temperature increase) or electron transfer to an unoccupied molecular orbital (LUMO) depending upon its relative position with respect to the Fermi level. Similarly, a reaction might be guided by the hole transfer between the HOMO of the adsorbate and metal[23,24]. In both cases, the reaction occurs in a non-thermal regime far from equilibrium where the transient molecular states undergo relaxation along their virtual potential energy surface leading to desorption, dissociation, or redox reactions within the femtosecond timescale[22]. Additionally, it has been shown that the HOMO-LUMO energy gap of the adsorbed molecules undergoes a shift compared to the free molecules in their metal-molecule hybrid states[25]. Also, the interfacial chemistry dictated by the presence of coadsorbed ions and ligands can influence the relaxation pathway and hence the reaction rate and product[26]. Likewise, the reaction efficiency can be ruled by photon density, energy, and external temperature[12,27]. Therefore, the aforementioned factors alone or altogether guide the reaction rate, selectivity, and yield of plasmonically driven surface reactions.

In this contribution, we aim to determine the factors contributing to the reactivity of a secondary amide molecule at the plasmonic interface of Au and Ag nanoparticles. Amides represent a pivotal binding motif not only present in peptides and proteins but also participating in common coupling reactions that are often used for nanoparticle functionalization[28]. Consequently, it is critical to understand its reactivity when located close to the surface of plasmonic nanoparticles under light irradiation. For this, we have chosen N-methyl-4-sulfanylbenzamide (NMSB) as a model molecule and studied its reactivity (under visible light irradiation) by surface-enhanced Raman scattering (SERS).

A commonly employed method in MS/MS analysis of peptides is electron capture dissociation (ECD) or electron transfer dissociation (ETD), enabling the fragmentation of peptides into smaller fragments useful in protein sequencing [29]. In both electron attachment processes, fragmentation

of peptides involves the loss of H atoms, backbone dissociations of N–C$_\alpha$ bonds, release of ammonia, side-chain groups, and disulfide bond cleavages. Therefore, we aimed to answer the following questions within this study: (a) What are the different plausible fragmentation products for plasmon-induced dissociative conversion of NMSB (secondary amide) in the nanocavity? (b) What is the influence of photon density, photon energy, external temperature, and plasmonic material on the reactivity of the NMSB in the plasmonic nanocavities? (c) What is the nature of the reactivity of NMSB in the nanocavities of Ag and Au aggregates under visible light irradiation? Based on this a plausible reaction mechanism will be drawn with the help of gas-phase DEA and density functional theory (DFT) studies. The overall dissociative conversion of the secondary amide (NMSB) is monitored in real-time using SERS, which provides a common platform to trigger and simultaneously track the respective reactions with a confocal Raman microscope[12].

## Results and discussion

We study the molecular transformation of NMSB adsorbed on self-aggregated structures of Ag and Au nanoparticles under plasmonic excitation with 633 nm and 785 nm laser sources. The SEM images in Fig. 1 show the morphological features of the NMSB-coated Ag (A and B) and Au (C) aggregates. The cartoon depiction in Fig. 1D (D1-D2) shows the chemical transformations occurring at the plasmonic nanocavities of Au and Ag. It has been reported before that plasmon-induced processes, and in particular hot-electron induced processes are strongly localized to the plasmonic nanocavities[12,30,31]. The reason is that the electric field enhancement is significantly higher in these nanocavities than at other parts of the nanoparticle surfaces. These are the locations where the transfer of hot electrons is the most probable. Hence we state that the reactions observed here are taking place in the hot spots formed by the nanoparticles.

SERS is used as a spectroscopic tool for the in-situ investigation to resolve the chemical transformation in terms of vibrational fingerprint during the reaction course. The overall spectral time evolution of the reaction is demonstrated in the SERS spectra shown in Fig. 2A–C recorded for Ag NP aggregates adsorbed with NMSB under 633 nm laser excitation. SERS spectra at different time intervals and time trajectories for the reaction of NMSB on Au aggregates under 633 nm laser excitations are shown in Supplementary Fig. 1.

**Fig. 2 | Overall time evolution of the transformation of a secondary amide tracked by SERS.**
**A** Representative SERS spectra obtained at different time intervals of the molecular transformation processes of NMSB on Ag aggregates (stack plots show the absolute intensities in each case); **B** Zoom-in SERS spectra to show the spectral change in **A** for the region 1100–1600 cm$^{-1}$; **C** Time-series SERS map recorded for 100 s under 633 nm laser excitation showing the transformation of NMSB molecule; **D** Reaction time traces of the peaks at 1320 cm$^{-1}$, 1189 cm$^{-1}$, and 2230 cm$^{-1}$ (extracted from time series map **C**) assigned to the loss of the amide III band, the appearance of the primary amide and the aromatic nitrile, respectively. **E** Reaction time traces of peaks at 315 cm$^{-1}$ and 345 cm$^{-1}$ assigned as NH$_2$ rocking vibration and C-S-Ag angle bending vibration, respectively. The evolution of these bands is attributed to the formation of two products viz., MBAm and MBN. The spectra have been recorded at 633 nm with a laser power of 2.7 mW and an acquisition time of 0.5 s.

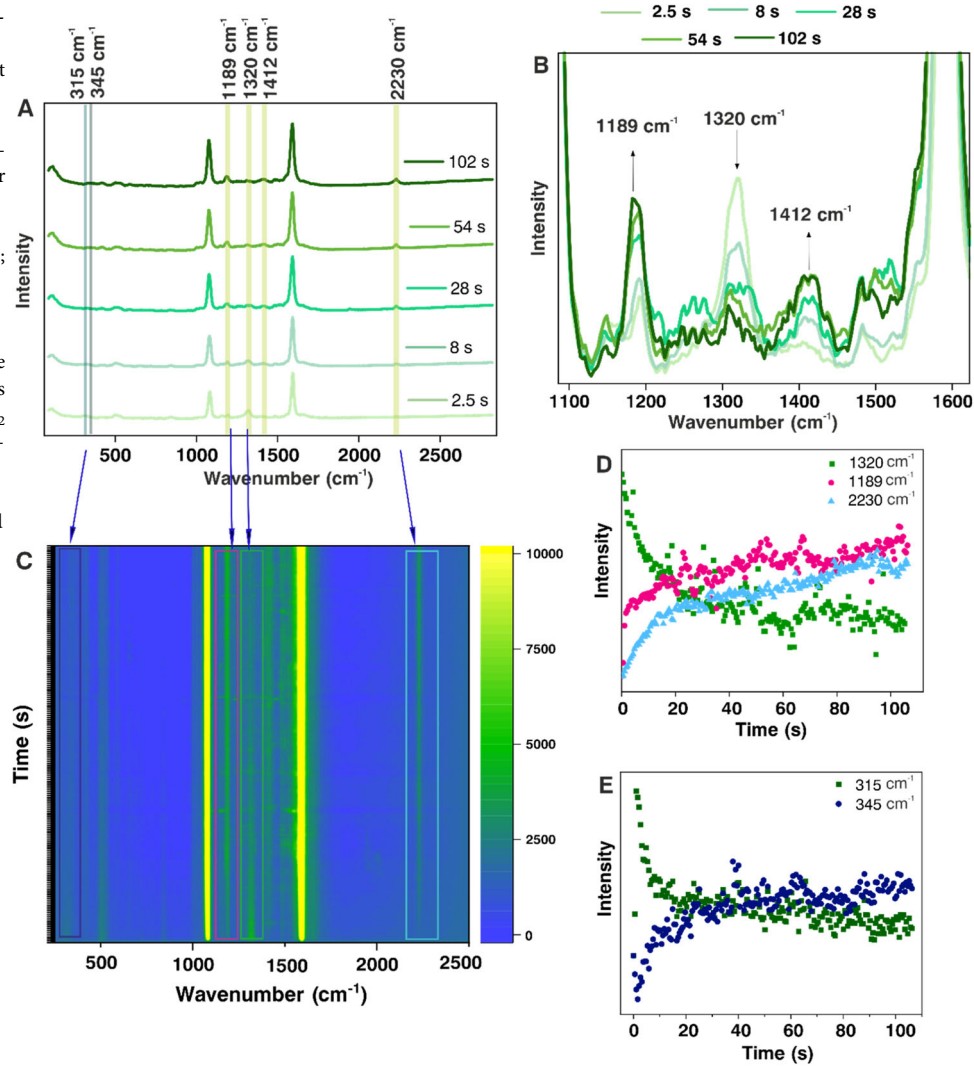

We considered the formation of different possible reaction products (molecular structures shown in Supplementary Fig. 2), namely Ag-S(C$_6$H$_4$)CONH$_2$, Ag-S(C$_6$H$_4$)CN, Ag-S(C$_6$H$_4$)CHO and Ag-S(C$_6$H$_4$)CO$^+$, which would show characteristic SERS peaks at 1189 cm$^{-1}$, 2230 cm$^{-1}$, 1659 cm$^{-1}$, 1756 cm$^{-1}$, respectively[14,32–34].

A clear gradual disappearance of the band at 1320 cm$^{-1}$ with time indicates the loss of the amide III band (secondary amide) during the reaction course[33]. This is accompanied by the appearance of the consistently rising peak at 1189 cm$^{-1}$, which is assigned to the NH$_2$ rocking vibration of a primary amide (Fig. 2B), indicating the transformation of NMSB to MBAm[14]. The additional peak arising at 1412 cm$^{-1}$ that corresponds to the amide III band of a primary amide further confirms the reaction product[32]. The reaction is accompanied by the formation of MBN indicated by the peak rise due to nitrile stretching vibration at 2230 cm$^{-1}$[34], (Fig. 2C, D) reported for the first time in the study.

The Raman spectra calculated for different possible reaction products are shown in Supplementary Figs. 3 and 4 considering M$_9$ clusters, M = Ag, Au, and M-S surface binding mode. The experimental SERS data shows peak characteristics of MBAm and MBN in a close match with that obtained from the calculated Raman spectral assignment. Other potential products are Ag-S(C$_6$H$_4$)CHO and Ag-S(C$_6$H$_4$)CO, respectively, which show characteristic peaks at ~1670 cm$^{-1}$ and ~1770 cm$^{-1}$ (Supplementary Fig. 3F, G) in the calculated Raman spectra. However, these were not observed in the SERS spectra indicating that such products have not been formed.

The overall transformation becomes also apparent by a peak shift from 315 cm$^{-1}$ to 345 cm$^{-1}$ during the reaction (Fig. 2E). To interpret the peak shift in the low wavenumber region, we compared the SERS spectra with the calculated Raman spectra obtained by DFT calculations shown in Fig. 3 and Supplementary Fig. 3. The stretching vibration at 311 cm$^{-1}$ (Fig. 3B) in the calculated Raman spectra considering Ag$_9$ clusters (details in the Methods section) has been assigned to NH$_2$ rocking vibration of MBAm whereas the one at 336 cm$^{-1}$ (Fig. 3C) corresponds to the C-S-Ag bending vibration in MBN.

This is in line with the observation made in SERS measurements and we infer the temporal blue shift observed from wavenumbers 315 cm$^{-1}$ to 345 cm$^{-1}$ to support the subsequent conversion of the primary product MBAm to MBN during the reaction course[35,36]. A relative red-shift (of the relevant bands in this study) in the SERS spectra for both Ag and Au could be seen when compared to the calculated Raman spectra in most of the cases. Although the spectral feature in the range 310–345 cm$^{-1}$ is quite broad, we refer for simplicity to the peak position at 315 cm$^{-1}$ and 345 cm$^{-1}$ to highlight the relative shift in the peak position during the reaction. The reaction under study is assumed to be irreversible because the reaction products are likely to diffuse away or even desorb, and hence a returnable reaction is not favorable in this case.

As introduced before, plasmon-induced chemical transformations are typically assigned to electron transfer and/or heat-induced processes. To further gain insights into the chemical transformation of NMSB at the plasmonic nanocavities, we also carried out a DEA study on the NMSB

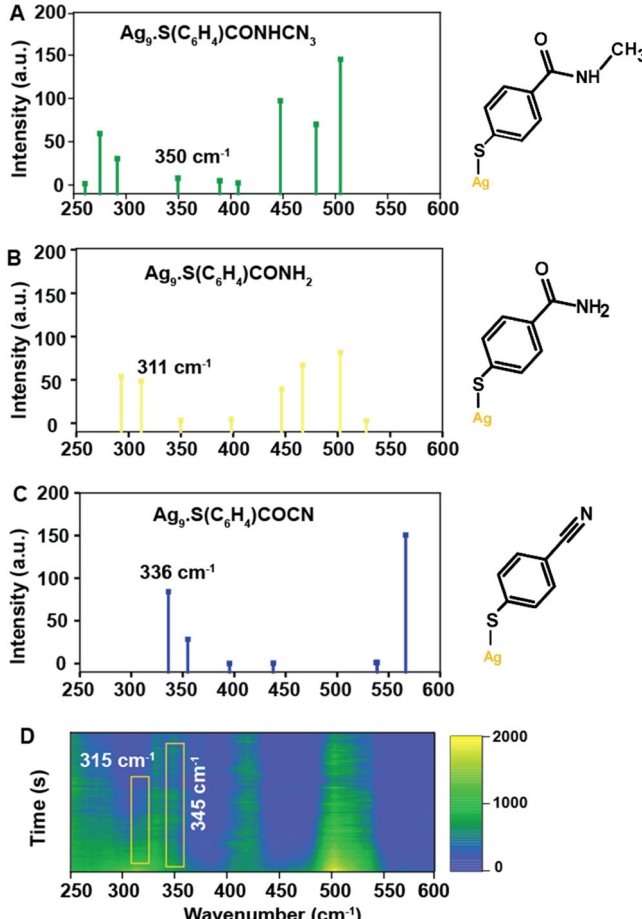

**Fig. 3 | Comparison of calculated Raman spectra and experimental SERS spectra.** Calculated Raman spectra (250–600 cm$^{-1}$) (**A**) NMSB, (**B**) MBAm and (**C**) MBN, compared with (**D**) experimental SERS spectra to support the peak assignment at 315 cm$^{-1}$ and 345 cm$^{-1}$ to NH$_2$ rocking vibration and C-S-Ag angle bending vibration of MBAm and MBN, respectively. Calculations were performed at the B3LYP/aug-cc-pVDZ level calculating the molecule adsorbed on an Ag$_9$ cluster (Supplementary Fig. 3).

molecule in the gas phase using mass spectrometry (see Methods for details). Previous studies on the reactivity of peptides reveal an electron attachment-mediated dissociation pathway via C$_\alpha$-N cleavage in the presence of a proton source studied by mass spectrometry[14]. In a similar context, low-energy electron attachment-induced bond dissociation of small peptides has also been reported in the gas phase[37]. Here, we experimentally observed four anionic fragments formed upon electron attachment to NMSB in the electron energy range from ~0 eV up to 12 eV (Fig. 4 and Supplementary Fig. 5, Supplementary Table 1). At low energies, hydrogen atom dissociation is observed, giving rise to [M–H]$^-$ at $m/z$ 166. The peak at low energies is considerably asymmetric, with a maximum at 1.1 eV, another peak is observed at 4.4 eV. The ion yield of [M–H]$^-$ is by about two orders of magnitude higher than all other yields. Another fragment anion is observed at $m/z$ 109, [M–C$_2$H$_4$NO]$^-$. The corresponding anion efficiency curve shows ion yield at energies above 4.3 eV, with two peak maxima at 5.7 eV and 8.6 eV. Finally, SH$^-$ and S$^-$ ions are observed at $m/z$ 33 and 32 at energies above 2.5 and 3.6 eV, respectively. The latter fragments are formed after passing over a [M–SH]…SH$^-$ pre-dissociation state; a detailed discussion about these fragment anions can be found in Supplementary Discussion 1. No ion yield indicating the relevant C$_\alpha$–N bond cleavage was observed within the detection limit of the used apparatus. This would correspond to the [M–CH$_3$]$^-$ fragment anion as well as the CH$_3^-$ fragment anion, for which it should be also noted that the electron affinity of CH$_3$ is very low (0.03 eV at

the CCSD(T)/aug-cc-pVTZ//CCSD/aug-cc-pVDZ level), and thus the formation probability of CH$_3^-$ could be low compared to other products. For the complementary reaction channel leading to [M–CH$_3$]$^-$, we computationally predict a threshold of 1.01 eV.

Therefore, the DEA studies in the gas phase do not provide direct evidence of C$_\alpha$–N cleavage of the NMSB molecule to prove the experimentally observed MBAm formation reflected in the SERS spectra, though the gas phase studies probe the action of low-energy electrons with similar kinetic energies like in the condensed phase experiment. It is known that plasmon excitation results in the generation of hot electrons in the range of 0–3 eV. Thus, we ascribe the discrepancy, when going from single molecules in the gas phase to molecules adsorbed on surfaces, to significantly change the DEA characteristics in terms of resonance energies, intensity of fragment ion formation, and/or specific dissociation pathways[38]. In detail, the main fragmentation reaction observed in the gas phase corresponds to S–H bond dissociation near electron energies of about 1.1 eV (Fig. 4 and Supplementary Fig. 5, Supplementary Table 1). Following a well-known mechanism in DEA[39,40], we can propose the capture of the electron into a π* orbital of the aromatic moiety, followed by coupling with the dissociative σ*(S–H) orbital.

The second reaction step is not possible to occur in the condensed phase since the sulfur atom is bound to the metal-nanoparticle. Instead, the electronic coupling of the π* state to the dissociative σ*(N–C$_\alpha$) state seems to become prevalent for condensed NMSB and thus would explain the observed N–C$_\alpha$ bond cleavage, absent in the gas phase experiment[40]. Other bond cleavages observed in the gas phase (like for example C–S bond) occur only at higher electron energies (Fig. 4) and will not play a considerable role in the plasmon experiment. Following the dissociation of the N–C$_\alpha$ bond, addition of a proton led to the formation of MBAm. Plausibly, water molecules trapped within the plasmonic substrate acted as a proton source albeit its energetically sluggish reaction kinetics (high water oxidative potential). This explains the reaction channels active in inducing the observed chemical transformations (although the detailed stepwise reactivity of the NMSB molecule is out of the scope of this study).

Additional mechanistic details can be obtained from SERS data recorded under various conditions. The reaction was also monitored under 785 nm excitation wavelength for Au and Ag NPs which showed a similar reaction trend (SERS spectra at different time intervals shown in Supplementary Figs. 6 and 7. That the reaction is not purely light-induced (i.e. without the involvement of plasmonic nanoparticles) is clear from our experimental observation where no reaction of the NMSB molecule deposited on Si-substrate (without nanoparticles) was observed at 633 nm and 785 nm laser excitation (Fig. 5A, B), indicating the role of plasmon excitation. Further, to understand the role of power and external temperature on the reactivity of NMSB molecule and to comment on the thermal or non-thermal nature of the reaction, we deduce the reaction rate constant for the dissociative transformation of NMSB molecule to MBAm. Considering the fast excitation and relaxation of hot-charge carriers under continuous wave (CW) illumination, the time-average concentration of the hot electrons is considered constant which allows us to consider the process to follow a pseudo-first-order rate law (see ref. 12 and Supplementary Discussion 2 for details) using the equation below:

$$\ln \frac{[\text{NMSB}]_0}{[\text{NMSB}]} = \frac{k_f}{1-h} \cdot t^{(1-h)} \qquad (1)$$

where [NMSB] and [NMSB]$_0$ represent the concentration of NMSB at time t and initial concentration, respectively. For kinetic calculation, the SERS intensity at a given wavenumber is considered proportional to the molecular concentration at a time "t". $k_f$ is the kinetic rate constant and h is a fractal term[41]. Relation (1) represents a power law function represented by $y = a \cdot t^b$ and therefore $k_f = (1-h) \cdot a = b \cdot a$. The time-series spectra obtained for loss of amide III peak at 1320 cm$^{-1}$ in all measurements are fitted to Eq. (1) to deduce the reaction rate constants

**Article**

**Fig. 4 | Dissociative electron attachment to the NMSB molecule.** Left: Anion efficiency curves for dissociative electron attachment to the NMSB molecule in the gas phase, producing [M–H]⁻, [M–C₂H₄NO]⁻, SH⁻, and S⁻ fragments. Right: Proposed ion structure and reaction energies (in eV) of forming the fragments from M + e⁻ as calculated at the CCSD(T)/aug-cc-pVDZ//ωB97XD/aug-cc-pVDZ level (see the SI for further details).

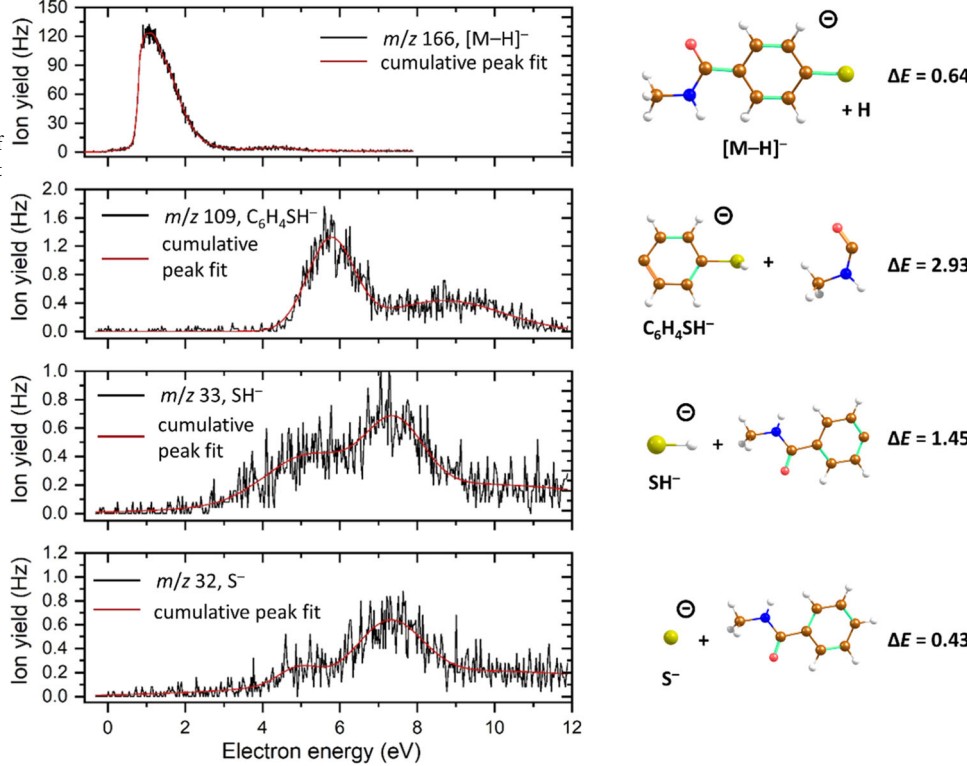

for all the kinetic measurements (at different laser power, wavelength, and substrate; Fig. 5C–F). A characteristic plot with the kinetic fit is shown in Supplementary Fig. 8. To further validate the contribution of plasmon in the molecular transformation of NMSB, we investigated the influence of laser power and external temperature on the reaction rate. A photochemical reaction is often marked by a linear increase of the reaction rate with increasing laser power, on the other hand, a thermal pathway could be predicted based on a super-linear dependence of the same[42]. We observed a near-linear dependence of the reaction rate constant on the laser power (calculated above) in the case of both Ag and Au under 633 nm, however, neither linear nor super-linear dependence of the reaction rate constant was observed in the case of 785 nm laser excitation (Fig. 5C, D). Additionally, hot-charge-driven reactions are often associated with thermal heating due to localized plasmon heating[43]. Therefore, a direct conclusion on the non-thermal nature of the dissociation reaction could not be deduced from Fig. 5C, D. This will be further substantiated in a later discussion of the study.

To be noted, Fig. 5D shows the comparison of three rate constant values calculated for Au aggregates under 785 nm excitation as no reaction could be observed at lower laser powers. Additionally, the reaction rate calculated under 633 nm laser excitation for both Ag and Au (Fig. 5C) is higher than that observed under 785 nm excitation (Fig. 5D). This can be attributed to the usually stronger plasmon absorbance band at 633 nm than at 785 nm for Ag and Au aggregate structures as shown in our previous micro absorbance study on 40 nm particle aggregates[12]. The UV-vis spectra recorded for AgNP dispersion coated with NMSB molecule (Supplementary Fig. 9) after centrifugal wash also substantiate the fact, although the UV-vis spectral feature reflects the situation for NMSB-treated particles in solution. The UV-vis spectra for NMSB-treated AuNPs dispersion is also shown in Supplementary Fig. 9B. Deconvolution of the UV-vis spectra (Supplementary Fig. 9A) shows a secondary plasmon band centered at 640 nm (approx.) which creates a near resonance with a 633 nm laser excitation than with a 785 nm excitation. This further supports the fact that the reaction rate depends on the availability of hot electrons (which is more favorable in the case of 633 nm than for 785 nm), Additionally, it was shown before that the

hot electron generation rate is higher for Ag than Au, which can account for the respectively higher reaction rates[44].

A schematic depicting the hot-electron attachment-mediated dissociative decomposition of NMSB at the plasmonic interface of Ag is shown in Scheme 1. The LUMO + 8 (Ag₉) and LUMO + 7 (Au₉) (Supplementary Fig. 10) represent the accessible ligand state molecular orbital for electron attachment respectively. The metal-molecule hybrid state representing either the metal or, ligand state molecular orbital when anchored on the surface of Au and Ag is shown in Supplementary Fig. 10.

Further, we conclude that the MBAm formed as the main product in this study further undergoes a reaction to generate MBN (although with a low yield, Fig. 2A, D) plausibly through a dehydration reaction channel. (Scheme 2) This is an important observation as it opens up a secondary reaction route for nitrile formation on the plasmonic surface under visible light irradiation.

A recent study on the dehydration reactions of a primary amide to nitrile at plasmonic junctions revealed the active and cooperative role of gold surface adatoms and hot electrons at the nanocavities[34]. With the help of the SERS study, Zhou et al. demonstrated that the gold adatoms at the plasmonic junction can form various metal-molecule complexes, stabilizing the reaction intermediates and facilitating the dehydration reaction. One may also speculate that the plasmonic heat could facilitate the low-yield MBN side product through a dehydration channel. In this context, the effect of external heat on the reactivity and reaction rate of NMSB was studied by monitoring the reaction at seven different temperatures between 25 °C and 85 °C (Fig. 5E, F). A slow rise in the reaction rate with increasing temperature was noted, most significant at 55 °C and 85 °C, for Au and Ag, respectively. However, the error bars in the case of both Au and Ag reflect a significant variation which does not allow us to conclude on a clear effect of temperature on the reaction rate. No significant effect of external heat on the dehydration reaction yield of MBN was observed. This further supports the less-dominant role of heat in the conversion of NMSB to MBAm and MBN. Therefore, we propose the predominance of hot-electron mediated DEA pathways in the chemical transformation of NMSB to MBAm and MBN on the surface of Ag and Au nanoparticles.

**Fig. 5 | Reactivity of NMSB molecule without nanoparticle and reaction rate study of the NMSB molecule with nanoparticle under different laser power and temperature conditions. A** Reaction time traces extracted from normal Raman spectra of NMSB (2.5 mM ethanolic solution) deposited on Si-substrate at 1320 cm$^{-1}$ and 1189 cm$^{-1}$ recorded under laser excitation 633 nm (7 mW) and (**B**) 785 nm (39 mW); Plot of reaction rate constant versus increasing laser power on the surface of Ag and Au under (**C**) 633 nm and (**D**) 785 nm laser excitation. Temperature dependence of the reaction rate constant on the surface of (**E**) Au (under 1.5 mW laser power) and (**F**) Ag (under 0.5 mW laser power) aggregates monitored at 633 nm laser illumination. The error bars represent the standard deviation of the rate constant values from three independent measurements.

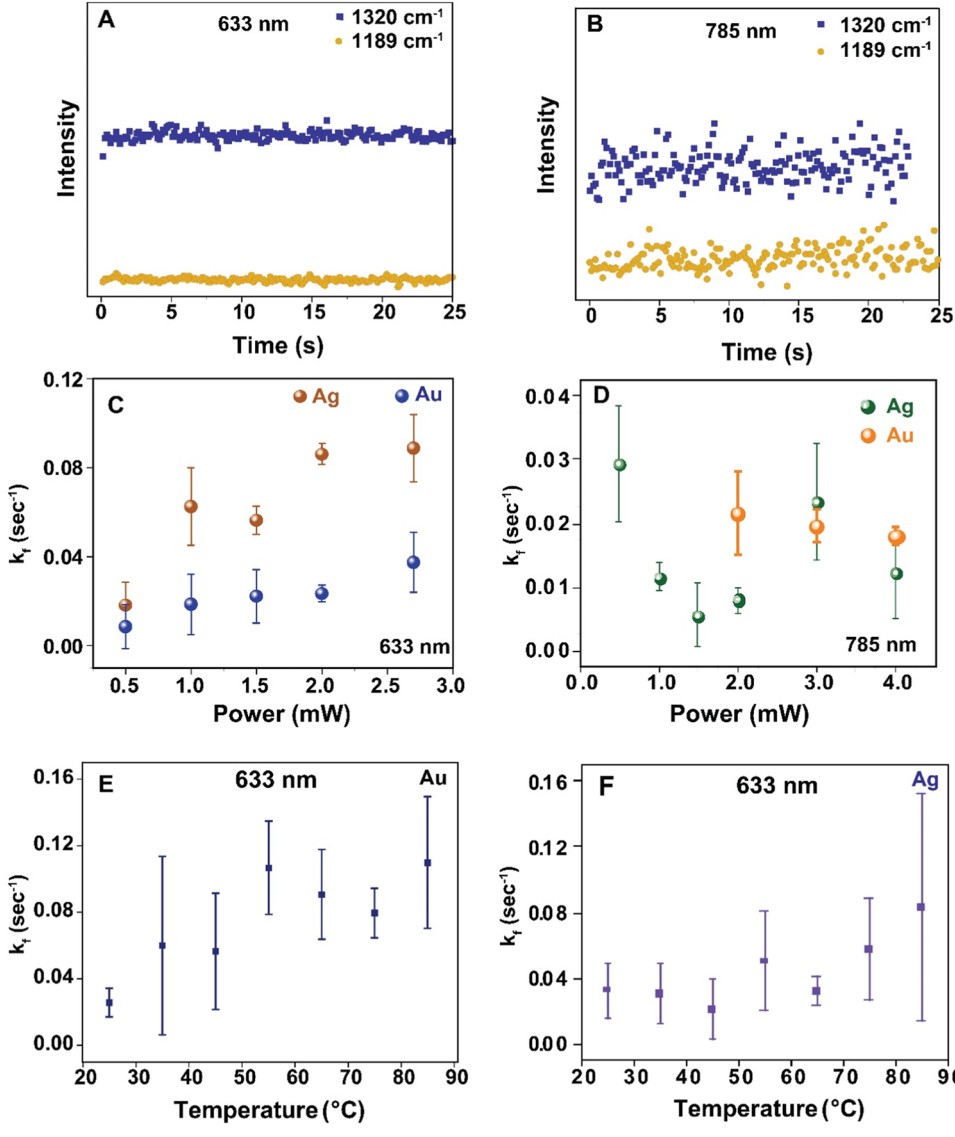

## Conclusions

Herein, we demonstrate the reactivity of a secondary amide (NMSB) under visible light irradiation in plasmonic nanocavities and develop an understanding of their reaction pathways by a SERS study. The characteristic NMSB peak disappearing at 1320 cm$^{-1}$ and new peaks rising at 1189 cm$^{-1}$ and 2230 cm$^{-1}$ confirmed the transformation of the secondary amide (NMSB) to a primary amide (MBAm), followed by the formation of an aromatic nitrile (MBN). The formation of MBN was confirmed by the appearance of C-S-M bending vibration in MBN (M = Ag and Au) during the chemical transformation and was further elucidated by DFT studies. Gas phase DEA studies reveal electron attachment to NMSB to form [M–H]$^-$ at electron energies below 2 eV. The electron captured in the $\pi^*$ orbital of the aromatic moiety can undergo electronic coupling with the dissociative $\sigma^*$(C-N) orbital, facilitating the N–C$_\alpha$ bond cleavage. Therefore, it is proposed that hot electrons generated under plasmonic excitation in the range of 1–3 eV can easily transfer this energy to the NMSB molecule to open up reaction channels via TNI states.

The reactions observed are not directly photo-induced, but require the presence of Au or Ag nanoparticles, which is verified by a constant signal from unaltered NMSB signal under direct 633 nm and 785 nm laser excitation when directly adsorbed on Si. A near-linear dependence of the reaction rate on the laser power under 633 nm laser excitation energy reflected the non-thermal nature of the reaction, however, no linearity was observed in the case of 785 nm laser excitation. Furthermore, despite significant error bars, a slow rise of the reaction rate with increasing temperature further substantiates the insignificant role of temperature in the reaction. Based on this observation, we conclude on a hot-electron mediated pathway for the observed chemical transformation, and that the reaction rate is limited by the availability of these. We believe that the conclusion drawn herein, based on the DEA studies and SERS kinetics, lays a strong ground to study a wide range of amides and peptides and their fragmentation pattern on plasmonic interfaces.

## Methods

### Chemicals

N-methyl-4-sulfanylbenzamide (NMSB; Sigma Aldrich and CHEM SPACE Co.; stated purity 95%), citrate AuNPs, and Ag NPs (diameter 40 nm) were purchased from Nanocomposix. Si-wafers were purchased from CrysTec GmbH (100 mm, p-Typ). All chemicals were used as received without further purification. Milli-Q grade water was used for all experimental preparations.

### Sample preparation

Citrate Ag and Au NPs were first functionalized with NMSB. 400 µL of as-synthesized AgNPs dispersion was purified by centrifugation at 7000 rcf for 8 min at 20 °C and the precipitate was redispersed in 320 µL of Milli-Q

water. The centrifuged AuNPs dispersion was mixed with 2.5 mM ethanolic solution of NMSB and left in a shaker for 2 h at 30 °C under 400 rpm for chemisorption on the NP's surface. The same steps were followed for sample preparation with AgNPs.

The resulting mixture was then centrifuged at 7000 rcf for 10 min at 20 °C (×2) and the precipitate was redispersed in 20 μL Milli-Q water. 10 μL of the sample dispersion was drop cast on cleaned Si-substrate and was left for drying at ambient conditions.

### Scanning electron microscopy (SEM) studies
SEM images were recorded with a cryo scanning electron microscope (Hitachi S-4800) under 2 kV accelerating voltage and 5.0 mm working distance for both Au and Ag NMSB aggregate samples.

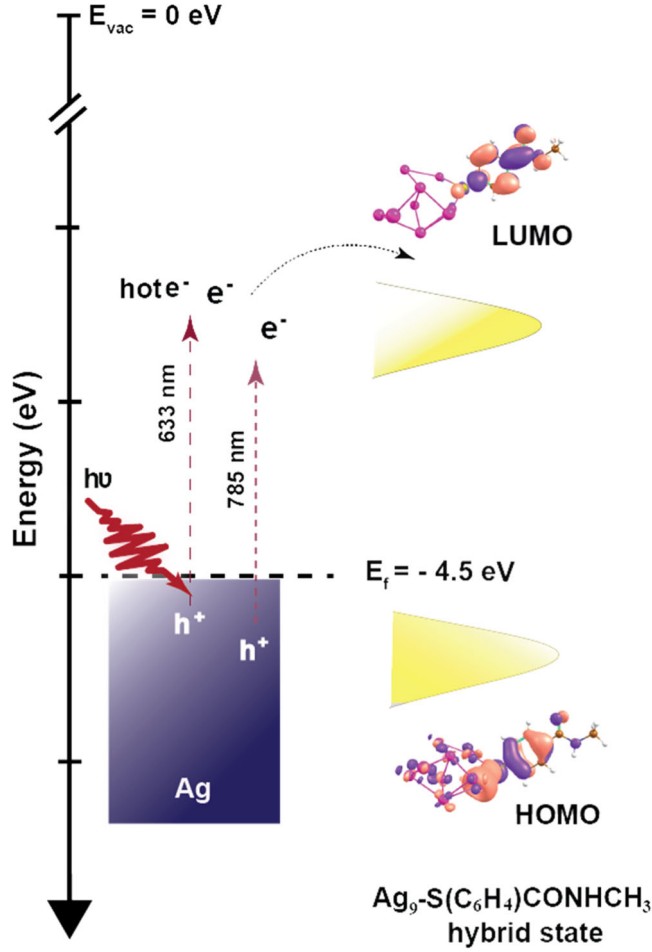

**Scheme 1 |** Schematic depicting the accessible HOMO-LUMO orbitals of the NMSB molecule when complexed with the metal surface (considering $Ag_9$ atoms) and the overall depiction of the hot-electron mediated chemical transformation of NMSB adsorbed on Ag nanostructure.

### SERS kinetic measurements
Kinetic measurements of NMSB-treated AuNPs and AgNPs drop cast on Si- Si-substrate were carried out using WITec alpha300 Raman – microscope. Two different excitation laser sources (632 nm, and 785 nm) were used for the measurements (100× objective; NA = 0.9). All measurements were carried out in triplicate in different sample areas. SERS spectra obtained were baseline corrected using WITec Project 5 and plotted using Origin 9.1 software. SERS spectra of bare nanoparticles are shown in Supplementary Fig. S11.

### Temperature-dependent studies
Similar steps were followed to prepare samples for temperature-dependent studies. The dispersion was drop cast on a smart substrate which are functionalized coverslips with an inbuilt transparent nanofabricated heating element and a temperature probe. The SERS studies were carried out directly on the smart substrate using a VAHEAT temperature controller from Interherence.

### Computations
Structures of gas-phase molecules and ions as well as molecules adsorbed on $Ag_9$ and $Au_9$ clusters were optimized at the $\omega$B97XD/aug-cc-pVDZ[45] and B3LYP/aug-cc-pVDZ levels. For gas-phase systems, a subsequent recalculation employing coupled clusters singles and doubles with non-iteratively included triples, CCSD(T)/aug-cc-pVDZ, was performed to obtain more reliable energies. The zero-point correction at the respective DFT level was applied for all reaction energies. To approach the vertical electron affinity, we used the aug-cc-pVTZ basis set with additional 2 s, 1 p and 1 d functions on each H atom, with basis function coefficients set as one-third of the ones present in the original basis set, further denoting this basis as aug-cc-pVTZ(H)+. In gas phase complexes, we found differences below 0.02 eV in CCSD(T) reaction energies for optimization at either DFT level. For both $Ag_9$ and $Au_9$, two cluster models with various adsorbed molecules were chosen to mimic the Raman spectra of the respective molecule on surfaces, with one and two bonds between sulfur and cluster surface. Here, the $\omega$B97XD functional predicts a strong interaction of the benzene ring and the C–O group with the cluster surface. Therefore, we use the B3LYP functional to obtain more reliable models of surface-molecule interactions. All systems were modeled in their lowest spin multiplicity, and wave function stabilization was performed before every calculation. All calculations were performed in the Gaussian program[46].

### Dissociative electron attachment (DEA) studies
A crossed electron-molecular beam instrument operated at high vacuum (~$10^{-8}$ mbar background pressure) was used. A more detailed description of the setup can be found in ref. 47. An oven with a capillary with a diameter of 1 mm was employed to introduce the molecules in the gas phase into the collision region with the electron beam. The electron source consisted of a homemade hemispherical electron monochromator (HEM). Ions formed were analyzed with a quadrupole mass analyzer. A channel electron multiplier was used for the detection of mass-selected ions. As a compromise between the resulting ion beam intensity and the electron energy resolution, an electron beam with an energy resolution of ~110 meV at FWHM

**Scheme 2 |** Schematic showing the proposed reaction steps for the formation of the reaction products under laser excitation at 633 nm and 785 nm on the surface of Au and Ag nanostructures.

(full width at half maximum) and a current of 40–45 nA was used. To obtain a reasonable ion signal without thermally decomposing the sample in the oven, the temperature dependency of the electron ionization mass spectrum was checked before starting measurements of negative ions. Measured with a Pt100 temperature sensor mounted directly to the oven, the compound was finally studied at a temperature of 353 K. At these conditions, the vacuum chamber pressure was $1.1 \times 10^{-7}$ mbar. For the calibration of the electron energy scale and determination of energy resolution, the well-known $Cl^-/CCl_4$ resonance at 0 eV was used[48].

## Data availability

The data that support the findings of this study are available from the corresponding author upon reasonable request. All data shown in the publication are archived at https://doi.org/10.5281/zenodo.13294167.

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

## Acknowledgements
AD thanks Sibylle Rüstig for recording the SEM images. IB acknowledges support by the European Research Council (ERC; consolidator Grant No. 772752) and the Deutsche Forschungsgemeinschaft (DFG, German Research Foundation) – CRC 1636 – Project ID 510943930 - Project No. A02. The computational results presented have been achieved using the HPC Infrastructure LEO of the University of Innsbruck.

## Author contributions
I.B. conceived and supervised the project. A.D. designed the project, performed all SERS experimental work, and interpreted the overall experimental data. M.O. performed quantum chemical calculations. S.D. supervised the DEA study and interpreted the DEA results. F.I. and E.A.-B. carried out the DEA measurements. F.I. analyzed the DEA data. J.A. assisted in the interpretation of the DEA experimental data. I.B., A.D., M.O., and S.D. contributed equally to writing the manuscript. All the authors have approved the final version of the manuscript.

## Funding

## Competing interests
The authors declare no competing interest.
