## [Peer Review File · Communications Chemistry]

Reviewers' comments:

Reviewer #1 (Remarks to the Author):

The paper called “Plasmon-Driven Chemical Transformation of a Secondary Amide Probed by Surface Enhanced Raman Scattering” by Dutta et al. deals with an interesting example of plasmon-assisted catalysis, as authors focused on a significant representative of amides, making their research relevant also from the biological/biochemical point of view. Even when I welcome the overall idea of the paper, I have encountered several places, where the attention of authors should be directed. The list of my comments is following:

Introduction

a) There are several issues with Figure 1. Firstly, what is the difference between figures A and B? According to the figure caption, they are both related to the AgNPs, while there is no significant difference between them. Is necessary to show both figures A and B, or better to say, does the presence of both figures bring some valuable information to the reader? If not, I would prefer to preserve only one of these figures.

Also, I have noticed that figures A/B (related to AgNPs) and C (related to the AuNPs) do not share the same scale bar. This could be confusing for the reader.

Also, more information about SEM images should be mentioned generally, for example, if the images were captured using backscattered or secondary electrons. This will be mentioned later again.

b) The sentence “Amides represent a pivotal binding motif not only” (starting on line 84) lacks a citation, as it carries significant information that has not originated from this paper.

c) The sentence “..., which provides a common platform to trigger and simultaneously track the reaction with a confocal Raman microscope” (ending on line 105) also seems to lack a citation, as this is not the first time when the Raman microscope was used for such purpose.

Results and Discussion

d) “The cartoon depiction in Fig. 1D (D1-D2) shows the chemical transformations occurring at the plasmonic nanocavities of Au and Ag.” (starting on line 110) – is there any convincing evidence that studied reactions are taking place only in the nanocavities? From my point of view, also molecules on the “top” of nanoparticles, meaning not trapped between them, would be susceptible to

undergo photochemical transformation. Nevertheless, this would be hard to state based only on the experiments, as the laser spot of both used lasers is much larger than these objects.

e) Figure 2 – B) Are the spectra shown in common scale or full scale? This should be stated, C) Frames (blue, purple, green, and so on) used to highlight spectral bands are barely visible – I see the effort to make their color matching to the ones used in Fig. 2 D and E, but at the current state, their color is too dark to be even noticed, D/E) Some comment about the vertical axis should be present – are trends plotted based on the absolute or normalized intensities of the bands? I would prefer to show the scale in this case, as the numbers can provide additional information about the speed of the ongoing processes. Also, it is not obvious if band areas or heights were used when plotting the trends – this should be clearly stated.

f) In the sentence “This is accompanied by the appearance of the consistently rising peak at 1189 cm^{-1} ,” (starting on line 131), the wavenumber does not match the one in Figure 2, where the wavenumber 1193 cm^{-1} is used.

g) It would be maybe useful to introduce also abbreviations of the potential products, not only for the starting molecule. This would make the whole text easier to read for the reader, who is not fully familiar with the studied structures. In general, the naming of the molecules could be united in the whole text, as during the text, sometimes full structures’ names, and sometimes their formulas are used, which forces the reader to return in the text in several places.

h) Wouldn’t it be useful to also plot the time evolution of the band at 1412 cm^{-1} ? As the authors claim that it’s arising confirms the reaction products, it seems to me that comparing the kinetics of this band and other related bands could bring a much more convincing argument.

i) Line 146 – The authors speak about the peak shift from 315 cm^{-1} to 345 cm^{-1} . Wouldn’t it be useful to also plot the peak position over time?

j) The position of the suggested secondary products’ peak is quite close to the position of the original NMSB peaks’ position (350 cm^{-1} according to the calculation). Authors should at least comment if there is a possibility of a returnable reaction on NMSB?

k) Line 238 – “That the reaction is not purely light-induced ...” – authors should provide more information about measurement without the NPs, i.e. what was the concentration of the deposited molecules, were they dissolved in the ethanolic solution too, and so on

l) Line 245 – “We consider a pseudo-first-order fractal rate law ...” authors should at least briefly comment, why such consideration is legible.

m) Equation (1) – it is not fully clear what authors used instead of concentration in this equation. I assume that the intensity of the SERS signal (this would correspond to the information in SI), but this must be clearly stated in the manuscript also.

n) Figure 5 – are the presented spectra SERS, or rather just Raman? Molecules are not deposited on the Ag/Au nanoparticles. D) The first three points for the Au are missing in the trend – I assume that this is because of the insufficient signal when using listed experimental conditions, but this should be briefly commented on. Also, the way of obtaining error bars should be commented on in the figure’s caption.

Conclusion

o) Line 328 – “That the reaction is not photo-induced is also supported ...” – reaction is definitely photo-induced. Please reconsider this sentence.

Methods

p) In the chapter “Chemicals”, the authors mentioned that they have used nanoparticles obtained from Nanocomposix. More detail should be provided, as Nanocomposix offers several types of 40 nm citrate nanoparticles for both Au and Ag.

q) In the chapter “Sample preparation” (line 349), there is the sentence “Similar steps were followed for sample preparation with AgNPs”. This would evoke in me that the preparation procedure was not the same for AuNPs and AgNPs. Was the preparation the same or just similar? If so, what were the differences between preparations?

r) In the chapter “SERS kinetic measurements” (line 355) there is the sentence “Four different excitation laser sources (632 nm, and 785 nm) were used for the measurements ...”. Please revise this sentence, as I assume that there was a small misunderstanding.

s) In the chapter “SERS kinetic measurements”, the authors should provide more detailed information about measurements. Was the signal collected in kinetic series, or as a set of individual measurements? Are the times in the figures associated with beginnings or with endings of the individual acquisitions? The authors speak about “processing using WITec Project 5 and Origin 9.1 softwares”. What procedures have spectra undergone? Baseline correction, averaging, ...? This must be clearly stated.

t) I am missing a chapter about SEM – at least some basic information, such as the used instrument, the value of the accelerating voltage, secondary/back-scattered electrons, and so on.

Supplementary information

u) Figure S1 – colors in Fig. 1S A are too similar, it would be better to use the same colors as in Figure 2, also, in Figures S1 B and C, I would prefer to show values on the vertical axis.

v) I wonder why there is no measurement of the Au-aggregates under 785 nm laser excitation included in the SI? Is there any specific reason for this? This could bring another valuable insight into the evaluation.

Apart from this, I have also several general comments:

I) I would like to see spectra of the bare nanoparticles, meaning deposited on the Si-wafer, but without molecules of NMSB. Based on my own experience, citrate-capped NPs could exhibit a non-negligible signal of the capping agent. It is possible that when using your experimental conditions, no signal of citrate molecules could be observed, but this must be confirmed to increase the credibility of the results.

II) Sadly, more deeper comparison of results obtained on Ag and Au aggregates is lacking. It is without any doubt that at least a comparison of the kinetics obtained when using the same excitation wavelength (e.g. rate of the reaction and so on) could bring another insight into the studied problematics.

III) Maybe, it would be useful to consider several reaction mechanisms, meaning differentiate between photocatalysis of molecular complexes (molecule-metal complexes should have red-shifted absorption band, as the authors themselves suggest in the sentence (line 71) "Additionally, it has been shown that the HOMO-LUMO energy gap of the adsorbed molecule undergoes a shift compared to the free molecule in their metal-molecule hybrid states". Authors should maybe comment option, that when using an excitation wavelength of 633 nm, studied processes could be assigned maybe rather to the "photochemistry of surface-complexes" than to the plasmon catalysis itself. This could be also the reason for the observation of differences when using an excitation wavelength of 785 nm because such excitation is probably positioned unfavorably according to the position of HOMO/LUMO orbitals of modified molecules (although this is also possible and was introduced recently, for example, on the molecules of amphetamines). Also, different positions of the plasmon resonance maxima (cited from the literature if it cannot be measured by the authors themselves) could be used for the argumentation.

Generally, I consider this paper well-written. Especially insight from the field of DEA and related DFT calculations is showing promising ability to aid when decoding observations introduced by Dutta et al. Nevertheless, as apparent from the comments listed above, the manuscript contains a non-negligible number of issues, which must be addressed before considering the acceptance of this paper in Communications Chemistry. However, I believe that when especially last three comments are satisfactorily answered, this paper could be fruitful for the readers of the Communications Chemistry. Because of that, I recommend asking authors for the major revision/resubmission of their manuscript.

Reviewer #2 (Remarks to the Author):

The manuscript entitled " Plasmon-Driven Chemical Transformation of a Secondary Amide Probed by Surface Enhanced Raman Scattering" (ID: COMMSCHEM-24-0113) investigated a model plasmon-mediated chemical reaction (PMCR) of a secondary amide using the time-resolved

surface-enhanced Raman spectroscopy (SERS), and provided the possible mechanisms of the phenomena during experiment. The study is interesting and of potential importance in the field of plasmonic catalysis. I would like to recommend acceptance for publication in Communications Chemistry after performing the following modifications.

1. The authors assert that the reaction is plasmon-driven, but they do not show the optical spectra of their system to prove there is a plasmon at 633 or 785 nm.
2. It would be beneficial for the authors to compare the differences in plasmon-driven reactions between Au nanoparticles (NPs) and Ag NPs in terms of their local surface plasmon resonance (LSPR) properties, hot electron energy, or other relevant factors.
3. The authors claim that the plasmon-driven reaction is initiated by hot-electrons rather than heat. Why external heat vs reaction rate can substantiate this claim? Why the influence of external heat is different for Au and Ag (Figure 5 E&F)?
4. Scheme 2 shows that the formation of p-mercaptobenzonitrile (MBN) is followed by the production of p-mercaptobenzamide (MBAm). This conclusion is drawn from Figure 2D (the SERS intensity time traces for $\nu(2230\text{ cm}^{-1})$ and $\nu(1193\text{ cm}^{-1})$), or from previous literature? Providing additional commentary on this point would enhance the clarity of the findings.
5. The arrangement of Figures can be improved. For example, the information provided from Figure 1a and 1b are similar.

The magnification of SEM images for Ag and Au aggregates should be the same for comparison.

The label position of D1 and D2 in Figure 1 is not appropriate.

It will be easier for readers if the author can label the reactant and products in D1 (NMSB) and D2 (e.g. MBN, MBAm).

The caption in Figure 2 only mentions the laser power, what's the laser spot area?

6. Carefully check the typos: For example, "binding motive"?

In Figure 2, the label is 1189 cm^{-1} in A and B, but the time traces in D become 1193 cm^{-1} .

"Four different excitation laser sources"  should be "Two different excitation laser sources"

The unit of temperature seems not the same in line 346 and 349?

Reviewer #3 (Remarks to the Author):

Plasmon-Driven Chemical Transformation of a Secondary Amide Probed by Surface Enhanced Raman Scattering

By: Anushree Dutta, Milan Ončák, Farhad Izadi, Eugene Arthur-Baidoo, João Ameixa, Stephan Denifl and Ilko Bald

Reviewers report.

The submitted work describes Plasmon driven conversion of the secondary amine N-methyl-4-sulfanylbenzamide (NMSB).

The time dependence and kinetics of the conversion is studied by surface enhanced Raman spectroscopy (SERS) and discussed in context to dissociative electron attachment (DEA) of the same molecule in the gas phase. The conversion and its kinetics are convincingly deduced from the time evolution of the observed Raman bands. The authors argue that the underlying reaction are mediated by electron transfer from the plasmonic nanostructures to the anchored NMSB molecules and argue that the underlying attachment process is the same in the gas phase, but the relaxation of the so formed transient negative ion (TNI) differs mainly due to absence of the dominating S-H dissociation channel when NMSB is anchored by the sulfur bond to the respective.

While the electron induced mediation of this reaction is apparent, the $\pi^* \sigma^*$ coupling mechanism leading to the -CH₃ dissociation is speculative but nonetheless plausible and a justified proposal.

Overall, the results are significant and of interest to the readers of Communications Chemistry, the data is at large of good quality and it's interpretation is plausible. However, the structure of the paper and individual paragraphs needs improvement before publication.

I recommend publication of the submission after the authors have considered the comment here below.

In general, rearrangement of the text as highlighted with "*" here below.

I also recommend that the authors give the manuscript a good read and attempt to simplify the sentence structure and refrain from ambiguous and general statements where these may be avoided. This applies also to the introduction though the points raised here below are limited to the results and discussion part.

Also, please check plural/singular and use of articles.

-Figure 2,caption:

The readability of the caption could be improved by using simple enumeration followed by the description of the respective panels instead of a mixture with flowing text, i.e., A): Description. B): Description...

-Figure 2:

The 1189 cm⁻¹ NH₂ rocking is shown at 1193 cm⁻¹ in in panel D. This is a bit confusing and not addressed in the text, while this vibration is noted at 1189 cm⁻¹ in the caption and the text.

Line 135

“In parallel, the reaction is accompanied by the subsequent formation of p-mercaptobenzonitrile...”

This is a bit a contradiction in terms, either the formation of p-mercaptobenzonitrile is parallel to that of p-mercaptobenzamide or subsequent. From the time dependence of the rise of the 2230 cm⁻¹ peak, I do not see any indication that the CN formation is subsequent to the diminishing 1320 cm⁻¹ or rise in 1189 cm⁻¹.

Please clarify this point.

Also, please consider to show the SER spectra with the calculated spectra in fig S3 (could then be shown as line spectra), for better comparison...the spectra of the intact NMSB in panel A and B and after exposure in the other panels. This would help the comparison.

Also, in panel A the hydrogen is missing from the sulphur in the formula.

And, the DFT MO's would better be placed in a separate figure. They do not contribute to the comparison of the calculated and measured spectra but make the figure crowded.

Figure 3:

Similar to Fig S3, for ease of comparison it would be good to show an overlay with the measured SARS in the respective panels, i.e. the intact NMSB in panel A and after extensive exposure in the other panels. A comparison with the color-coded time dependence is not very clear.

Line 170...

“...blue shift observed from wavenumbers 315 cm⁻¹ to 345 cm⁻¹ to support the subsequent conversion of p-mercaptobenzamide to p-mercaptobenzonitrile during the reaction course.”

Please clarify what is meant by subsequent conversion...is this a secondary DEA process...is that reflected in the kinetics?

Line 201

“...of the NMSB molecule to prove the experimentally observed p-mercaptobenzamide formation...”

Propose to reformulate this sentence to read:

“...of the NMSB molecule to further substantiate the p-mercaptobenzamide formation...”

Lines 231-232:

“Concurrently, the water molecules trapped within the plasmonic substrate can act as a proton source albeit its energetically sluggish reaction kinetics to undergo the subsequent transformation to form p-mercaptobenzamide.”

This sentence needs to be rewritten and given some substantiated meaning. i) it is not clear what event this one is concurrent to, ii) the water molecules trapped within the plasmonic substrate have not been discussed...is this common knowledge, iii) what is “energetically sluggish reaction kinetics” and iv) it is not clear what the transformation to form p-mercaptobenzamide is subsequent to.

The following sentence should also be revisited.

“This explains the plausible reaction channels active in our case to induce the observed chemical transformations (although the detailed stepwise reactivity of the NMSB molecule is out of the scope of this study).”

This is a very general statement and apparently withdrawn within the concluding parenthesis. Please rephrase to clarify what reaction channels the sentence above “explains” and substantiate or put forward the causality in a clear way.

264 dependency -> dependence

Line 238...:

“That the reaction is not purely light-induced (i.e. without involvement of plasmonic nanoparticles) is clear from our experimental observation where no reaction of the NMSB molecule deposited on Si-substrate (without nanoparticles) was observed at 633 nm and 785 nm laser excitation (Fig. 5A, B) indicating the role of plasmon excitation. Further, to understand the role of power and external temperature on the reactivity of NMSB molecule and to comment on the thermal or non-thermal nature of the reaction, we deduce the reaction rate constant for the dissociative transformation of NMSB molecule to p-mercaptobenzamide.”

*I recommend to place this before the discussion of the 633 nm data as it applies equally at both wave lengths.

*In fact I believe the paper would benefit from being rearranged such that all the plasmon driven reactions, including the kinetics are discussed before turning to the DEA data discussion. It is distracting and does not benefit the structure or coherence to split the plasmon driven chemistry up like this.

Figure 5.

Please state clearly in the caption that panels A and B are from a Si substrate and if possible, signify that in the panels. Also please be consistent with the terminology. The intensity vs time spectra are termed “Time-lapse SERS trajectory”, “The time evolution of the peaks” and “time trajectories”...maybe more...

Figure 5,caption

The readability of the caption could be improved by using simple enumeration followed by the description of the respective panels instead of a mixture with flowing text, i.e., A): Description. B): Description...

Also clarify what those mW are referring to, and if this is the laser power, please also state that in other places where it is of relevance.

Line 274... :

“A schematic depicting the hot-electron transfer mediated dissociative decomposition of NMSB at the plasmonic interface of Ag is shown in Scheme 1. The accessible HOMO-LUMO orbitals of the NMSB molecule when anchored on the surface of Au and Ag are shown in Fig. S8.”

Please extend the discussion on scheme 1 to bring it into context with the discussion of the DEA process, and please clarify the relationship to the MOs shown in Fig S8. In S8 there are no orbital lobes on NMSB except for LUMO +8 (Ag) and LUMO +7 (Au).

Lines 289 -292

“Additionally, we observe that the p-mercaptobenzamide formed as the main product in this study further undergoes a reaction to generate p-mercaptobenzonitrile (although with a low yield, Figure 2A and D) plausibly through a dehydration reaction channel. (Scheme 2) This is an important observation as it opens up a secondary reaction route for nitrile formation on the plasmonic surface under visible light irradiation.”

This is partly the answer to few of my comments here above and needs to be placed in context to these claims when they are put forward...at which time they appear unclear and not substantiated. At least, though late, here a hypothesis is offered for the subsequent formation of the nitrile.

Line 303

“In this context, the effect of external heat on the reactivity and reaction rate of NMSB was studied by monitoring the reaction at seven different temperatures between 25 °C and 85 °C (Fig. 5E and F). A slow increment in the reaction rate with increasing temperature was noted in the case of both Au and Ag under 633 nm laser excitation. No profound effect of external heat on the dehydration reaction yield of p-mercaptobenzonitrile was noted. This further evidenced the less-dominant role of heat in the conversion of NMSB to p-mercaptobenzamide and p-mercaptobenzonitrile”.

Looking at Fig. 5E and F, there is an apparent rise in k above 50 Deg C in both cases and at the Au particles, k has increased more than three fold (about 2x on Ag). The error bars are however significant and I believe that the only conclusion that may be drawn here is that no significant temperature effect may be deduced from this data.

Reviewer #1 (Remarks to the Author):

The paper called “Plasmon-Driven Chemical Transformation of a Secondary Amide Probed by Surface Enhanced Raman Scattering” by Dutta et al. deals with an interesting example of plasmon-assisted catalysis, as authors focused on a significant representative of amides, making their research relevant also from the biological/biochemical point of view. Even when I welcome the overall idea of the paper, I have encountered several places, where the attention of authors should be directed. The list of my comments is following:

Introduction

a) There are several issues with Figure 1. Firstly, what is the difference between figures A and B? According to the figure caption, they are both related to the AgNPs, while there is no significant difference between them. Is necessary to show both figures A and B, or better to say, does the presence of both figures bring some valuable information to the reader? If not, I would prefer to preserve only one of these figures.

COMMENT: We thank the reviewer for taking a closer look at Figure 1. We identified the problem and have corrected it in the revised manuscript. (Page 4)

Figures 1A and 1C represent the SEM image of Ag and Au respectively. Figure 1B (shown for AgNPs coated with NMSB) is shown to highlight the honeycomb-like arrangement of the respective aggregates.

Also, I have noticed that figures A/B (related to AgNPs) and C (related to the AuNPs) do not share the same scale bar. This could be confusing for the reader. Also, more information about SEM images should be mentioned generally, for example, if the images were captured using backscattered or secondary electrons. This will be mentioned later again.

COMMENT: The respective Figures were not labeled correctly, which led to confusion. This has been corrected now in the revised manuscript. (Page-4) Figure 1A and 1C shows NMSB coated aggregated Ag and AuNPs (shown in same scale bar) for comparison. Figure 1B is shown to highlight the honey comb arrangement where NMSB coated aggregated Ag is shown (at a different scale bar) Additional information relating to SEM image acquisition and instrument specifications have been added.

b) The sentence “Amides represent a pivotal binding motif not only ...” (starting on line 84) lacks a citation, as it carries significant information that has not originated from this paper.

COMMENT: Incorporated in the revised manuscript as reference 28.

c) The sentence “..., which provides a common platform to trigger and simultaneously track the reaction with a confocal Raman microscope” (ending on line 105) also seems to lack a citation, as this is not the first time when the Raman microscope was used for such purpose.

COMMENT: Incorporated in the revised manuscript as reference 12.

“The overall dissociative conversion of secondary amide (NMSB) is monitored in real-time using SERS, which provides a common platform to trigger and simultaneously track the reaction with a confocal Raman microscope.¹²”

Results and Discussions

d) “The cartoon depiction in Fig. 1D (D1-D2) shows the chemical transformations occurring at the plasmonic nanocavities of Au and Ag.” (starting on line 110) – is there any convincing evidence that studied reactions are taking place only in the nanocavities? From my point of view, also molecules on the “top” of nanoparticles, meaning not trapped between them, would be susceptible to undergo photochemical transformation. Nevertheless, this would be hard to state based only on the experiments, as the laser spot of both used lasers is much larger than these objects.

COMMENT: It has been shown before that plasmon-induced processes, and in particular hot-electron induced processes are strongly localized to the plasmonic cavities. The reason is that the electric field enhancement is significantly higher in these nanocavities than at the other parts of the nanoparticle surfaces. These are the locations where the transfer of hot electrons is the most probable. Consequently, we assume that the reactions observed here are taking place in these hot spots formed by the nanoparticles because it is the most probable scenario. Nevertheless, if the plasmon-induced processes are mainly driven by an increased temperature, then the reactions can be less localized because the temperature increase might involve the nanoparticles as a whole. However, we don't have indications that for the present system and reactions we have a strong influence of temperature on the reaction outcome as is discussed in

the manuscript. Additionally, the SERS signal used to monitor the reactions is generated only from the nanocavities, therefore we are only probing the nanocavities. In a first approximation, we assume that the locations of highest SERS signal are also the locations of highest reaction rates. Therefore, emphasis has been laid on the reaction that takes place at the nanocavities in the manuscript. This has been well discussed in the previous reports by various groups -

For the clarity of the readers, we have added the following lines in the revised manuscript:

“It has been reported before that plasmon-induced processes, and in particular hot-electron induced processes are strongly localized to the plasmonic nanocavities. The reason is that the electric field enhancement is significantly higher in these nanocavities than at the other parts of the nanoparticle surfaces. These are the locations where the transfer of hot electrons is the most probable. Consequently, we assume that the reactions observed here are taking place in these hot spots formed by the nanoparticles.”

e) Figure 2 – B) Are the spectra shown in common scale or full scale? This should be stated, **C)** Frames (blue, purple, green, and so on) used to highlight spectral bands are barely visible – I see the effort to make their color matching to the ones used in Fig. 2 D and E, but at the current state, their color is too dark to be even noticed, **D/E)** Some comment about the vertical axis should be present – are trends plotted based on the absolute or normalized intensities of the bands? I would prefer to show the scale in this case, as the numbers can provide additional information about the speed of the ongoing processes. Also, it is not obvious if band areas or heights were used when plotting the trends – this should be clearly stated.

***COMMENT:** The spectra in Figure 2B represents the zoom-in spectra of Figure 2A., It is stated in Figure caption 2B in the revised manuscript.*

We have considered the SERS signal intensity in all the cases to showcase the plot trends.

f) In the sentence “This is accompanied by the appearance of the consistently rising peak at 1189 cm⁻¹,” (starting on line 131), the wavenumber does not match the one in Figure 2, where the wavenumber 1193 cm⁻¹ is used.

***COMMENT:** We thank the reviewer for noticing the error in Figure 2D. It was a typo and it is now corrected in Figure 2D in the revised manuscript.*

g) It would be maybe useful to introduce also abbreviations of the potential products, not only for the starting molecule. This would make the whole text easier to read for the reader, who is not fully familiar with the studied structures. In general, the naming of the molecules could be united in the whole text, as during the text, sometimes full structures' names, and sometimes their formulas are used, which forces the reader to return to the text in several places.

COMMENT: Necessary changes have been incorporated in the revised manuscript.

h) Wouldn't it be useful to also plot the time evolution of the band at 1412 cm⁻¹? As the authors claim that it confirms the reaction products, it seems to me that comparing the kinetics of this band and other related bands could bring a much more convincing argument.

COMMENT: Indeed following the kinetics using the evolution of the band at 1412 cm⁻¹ would have been useful, based on the rate law concerning the product formation. However, this requires the knowledge of the detailed reaction mechanistic steps taking into consideration the hole mechanism, by-products formed, etc. which makes the rate law deduction with respect to product formation very complex. Hence we preferred to deduce the reaction kinetics based on the more straightforward reactant decay.

i) Line 146 – The authors speak about the peak shift from 315 cm⁻¹ to 345 cm⁻¹. Wouldn't it be useful to also plot the peak position over time?

COMMENT: We believe that the above two suggestions (h and i) could be added as supporting data for the manuscript but necessarily do not bring forward new information to draw a more convincing conclusion.

j) The position of the suggested secondary products' peak is quite close to the position of the original NMSB peaks' position (350 cm⁻¹ according to the calculation). Authors should at least comment if a returnable reaction is possible on NMSB.

COMMENT: The reaction under study is assumed to be irreversible because the reaction products are likely to diffuse away or even desorb, and hence a returnable reaction is not favorable in this case.

k) Line 238 – “That the reaction is not purely light-induced ...” – authors should provide more

information about measurement without the NPs, i.e. what was the concentration of the deposited molecules, were they dissolved in the ethanolic solution too, and so on.

COMMENT: We thank the reviewer for this question: The concentration of the ethanolic solution (2.5 mM) used for the control measurement in the study has been incorporated into the caption of Figure 5 of the revised manuscript.

l) Line 245 – “We consider a pseudo-first-order fractal rate law ...” authors should at least briefly comment, on why such consideration is legible.

COMMENT: The explanation is provided in the SI as stated below:

We have stated the general rate equation of the reaction under study as follows:

$$\text{Rate} = k \cdot [\text{NMSB}] [e^-]$$

Considering fast excitation and relaxation of hot-charge carriers under continuous wave (CW) illumination, the time-average concentration of the hot electrons is considered constant which allows us to consider the process to follow a pseudo-first-order rate law.

Further, the explanation for the fractal rate law kinetics was already stated in the manuscript which is as follows:

“We take into account a fractal kinetic term in the rate equation to include the inhomogeneity of the substrate.(find similar discussion in ref 12) The inhomogeneous distribution of the reaction site induces a time dependence of the electron transfer and hence the rate constant, which is taken care of by considering the fractal term expressed by equation (S5)2

$$k_1 = k_f \cdot t^{-h} , 0 \leq h \leq 1 \text{ and } t \geq 1 \quad (\text{S5})$$

where k_1 refers to the time-dependent reaction rate constant, k_f is the kinetic corrected time-independent rate constant and will be used in the kinetic analysis to deduce the reaction rate constant and h represents the kinetic fractal term.³”

m) Equation (1) – it is not fully clear what authors used instead of concentration in this equation. I assume that the intensity of the SERS signal (this would correspond to the information in SI), but this must be clearly stated in the manuscript also.

COMMENT: *For the kinetic fit, the SERS intensity is considered proportional to molecular concentration at a particular time “t”.*

Therefore, the following line has been added to the revised manuscript, (Page 13)

“For kinetic calculation, the SERS intensity at a particular time for a given wavenumber is considered proportional to molecular concentration.”

n) Figure 5 – are the presented spectra SERS, or rather just Raman? Molecules are not deposited on the Ag/Au nanoparticles. D) The first three points for the Au are missing in the trend – I assume that this is because of the insufficient signal when using listed experimental conditions, but this should be briefly commented on. Also, the way of obtaining error bars should be commented on in the figure’s caption.

COMMENT: *The Figure 5 caption has been rephrased and the necessary information has been incorporated in the revised manuscript. (Page 14)*

“To be noted, Figure 5D shows the comparison of three rate constant values calculated for Au aggregates under 785 nm excitation as no reaction could be observed at the lower laser powers.”

Conclusion

o) Line 328 – “That the reaction is not photo-induced is also supported ...” – reaction is photo-induced. Please reconsider this sentence.

COMMENT: *We apologize for the confusion. We wanted to state that the reaction is not purely photoinduced, i.e. in the absence of nanoparticles. This is evident from the control experiment shown in Figures 5A and B. Time-lapse spectra showed no change in the Raman intensity of the respective reactant and product peak of the NMSB molecule. This is now phrased more clearly in the conclusions.*

“The reaction is not purely photo-induced (i.e. it requires the presence of nanoparticles), which is suggested by a constant signal due to the NMSB molecule under direct 633 nm and 785 nm laser excitations.”

Methods

p) In the chapter “Chemicals”, the authors mentioned that they have used nanoparticles obtained from Nanocomposix. More detail should be provided, as Nanocomposix offers several types of 40 nm citrate nanoparticles for both Au and Ag.

COMMENT: The nanoparticles were spherical citrate stabilized Au and Ag particles with 40 nm diameter. This is now stated in the text.

q) In the chapter “Sample preparation” (line 349), there is the sentence “Similar steps were followed for sample preparation with AgNPs”. This would evoke in me that the preparation procedure was not the same for AuNPs and AgNPs. Was the preparation the same or just similar? If so, what were the differences between preparations?

COMMENT: The sentence has been rephrased now in the revised manuscript.

r) In the chapter “SERS kinetic measurements” (line 355) there is the sentence “Four different excitation laser sources (632 nm, and 785 nm) were used for the measurements ...”. Please revise this sentence, as I assume that there was a small misunderstanding.

COMMENT: We thank the reviewer for the correction and the changes have been incorporated in the revised manuscript.

s) In the chapter “SERS kinetic measurements”, the authors should provide more detailed information about measurements. Was the signal collected in kinetic series, or as a set of individual measurements? Are the times in the figures associated with beginnings or with endings of the individual acquisitions? The authors speak about “processing using WITec Project 5 and Origin 9.1 software”. What procedures have spectra undergone? Baseline correction, averaging, ...? This must be clearly stated.

COMMENT: The SERS spectra were collected in kinetic series and do not represent individual measurements. The time series spectral data used for kinetic calculations were baseline-corrected using WITec Project 5 and plotted using Origin 9.1 software.

Additional information has been incorporated in the revised manuscript.

t) I am missing a chapter about SEM – at least some basic information, such as the used

instrument, the value of the accelerating voltage, secondary/back-scattered electrons, and so on.

COMMENT: The following information has been added to the manuscript:

“Scanning electron microscopy (SEM) studies. SEM images were recorded with a cryo scanning electron microscope (Hitachi S-4800) under 2kV accelerating voltage and 5.0 mm working distance for both Au and Ag NMSB aggregate samples.”

Supplementary information

u) Figure S1 – colors in Fig. 1S A are too similar, it would be better to use the same colors as in Figure 2, also, in Figures S1 B and C, I would prefer to show values on the vertical axis.

COMMENT: Necessary changes have been incorporated in Figure S1.

v) I wonder why there is no measurement of the Au-aggregates under 785 nm laser excitation included in the SI. Is there any specific reason for this? This could bring another valuable insight into the evaluation.

COMMENT: The SERS spectra at different time intervals of NMSB reactivity on Au aggregates under 785 nm is shown in Figure S7 which shows a similar trend as observed with 633 nm.

Apart from this, I have also several general comments:

D) I would like to see spectra of the bare nanoparticles, meaning deposited on the Si-wafer, but without molecules of NMSB. Based on my own experience, citrate-capped NPs could exhibit a non-negligible signal of the capping agent. It is possible that when using your experimental conditions, no signal of citrate molecules could be observed, but this must be confirmed to increase the credibility of the results.

COMMENT: The SERS spectra recorded on bare aggregated 40 nm citrate capped Au NPs and Ag NPs (633nm laser, laser power - 500uW, acquisition time - 50 ms) are shown below. The SERS spectrum of NMSB-coated Ag NPs is shown for reference.

II) Sadly, more deeper comparison of results obtained on Ag and Au aggregates is lacking. It is without any doubt that at least a comparison of the kinetics obtained when using the same excitation wavelength (e.g. rate of the reaction and so on) could bring another insight into the studied problematics.

COMMENT: *Additional discussions have been incorporated in the revised manuscript. Page 14,*

“To be noted, Fig. 5D shows the comparison of three rate constant values calculated for Au aggregates under 785 nm excitation as no reaction could be observed at the lower laser powers. Additionally, the reaction rate calculated under 633 nm laser excitation for both Ag and Au (Fig. 5C) is higher than that observed under 785 nm excitation (Fig. 5D). This can be attributed to the usually stronger plasmon absorbance band (reflected from our previous micro absorbance study on 40 nm particle aggregates) at 633 nm than at 785 nm for Ag and Au aggregate structures relevant for our study.¹² The UV-vis spectra recorded for Ag dispersion coated with NMSB molecule after centrifugal wash also substantiate the fact, although the UV-vis spectral feature reflects the situation for NMSB-treated particles in solution. The UV-vis spectra for NMSB-treated AuNPs are also shown in Fig. S9B. Deconvolution of the UV-vis spectra (Fig.S9A) shows a secondary plasmon band centered at 640 nm (approx.) which creates a near resonance with a 633 nm laser excitation than a 785 nm source. This is indicative of the fact that the reaction rate depends on the availability of hot electrons (which is more favorable in the case of 633 nm than for 785 nm) and it was shown before that the hot electron generation rate is higher for Ag than Au, which can account for the higher reaction rates.⁴³”

III) Maybe, it would be useful to consider several reaction mechanisms, meaning differentiate between photocatalysis of molecular complexes (molecule-metal complexes should have red-shifted absorption band, as the authors themselves suggest in the sentence (line 71) “Additionally, it has been shown that the HOMO-LUMO energy gap of the adsorbed molecule undergoes a shift compared to the free molecule in their metal-molecule hybrid states”. Authors should maybe comment option, that when using an excitation wavelength of 633 nm, studied processes could be assigned rather to the “photochemistry of surface-complexes” than to the plasmon catalysis itself. This could be also the reason for the observation of differences when using an excitation wavelength of 785 nm because such excitation is probably positioned unfavorably according to the position of HOMO/LUMO orbitals of modified molecules (although this is also possible and was introduced recently, for example, on the molecules of amphetamines). Also, different positions of the plasmon resonance maxima (cited from the literature if it cannot be measured by the authors themselves) could be used for the argumentation.

COMMENT: *We thank the reviewer for their deeper thoughts on understanding the probable reaction mechanism for the studied reaction system. Indeed we have considered the photocatalysis mediated by the metal-molecule complex in our study. HOMO-LUMO states in the metal-molecule complex obtained by DFT calculation (shown in Figure below) showed a shift compared to the free molecule. However, the excitation energy corresponding to the 633 nm laser is insufficient to induce the possible light-induced change due to the unfavorable HOMO-LUMO position of modified molecules. In addition, we see that the molecule is inactive under visible light excitation (Figure 5 A and B, using 633 nm and 785 nm). Therefore, we rule out the possibility of metal-molecule complex-induced photocatalysis and introduce the role of plasmons in driving the reaction.*

The role of plasmon resonance maxima has been considered in the revised manuscript and a necessary argument has been added (Page 14)

Generally, I consider this paper well-written. Especially insight from the field of DEA and related DFT calculations is showing promising ability to aid when decoding observations introduced by Dutta et al. Nevertheless, as apparent from the comments listed above, the manuscript contains a non-negligible number of issues, which must be addressed before considering the acceptance of this paper in Communications Chemistry. However, I believe that when especially last three comments are satisfactorily answered, this paper could be fruitful for the readers of Communications Chemistry. Because of that, I recommend asking authors for the major revision/resubmission of their manuscript.

Reviewer #2 (Remarks to the Author):

The manuscript entitled " Plasmon-Driven Chemical Transformation of a Secondary Amide Probed by Surface Enhanced Raman Scattering" (ID: COMMSCHEM-24-0113) investigated a model plasmon-mediated chemical reaction (PMCR) of a secondary amide using the time-resolved surface-enhanced Raman spectroscopy (SERS), and provided the possible mechanisms of the phenomena during the experiment. The study is interesting and of potential importance in the field of plasmonic catalysis. I would like to recommend acceptance for publication in Communications Chemistry after performing the following modifications.

1. The authors assert that the reaction is plasmon-driven, but they do not show the optical spectra of their system to prove there is a plasmon at 633 or 785 nm.

COMMENT: *The UV-vis spectra of NMSB-coated Ag NPs and AuNPs obtained after centrifugal wash are presented in the Supporting Information (Figure S9). Although deconvolution of the UV-vis band, shows a secondary plasmon band in case of Ag NPs treated with NMSB, however, AuNPs do not show any secondary plasmon band. However, this reflects the situation for NMSB treated particles in solution.*

Aggregated particles (40 nm) on a solid substrate (relevant to our study) show very broad and almost structureless absorption within a broad spectral range including 633 nm and 785 nm (please refer to ref. 12 of the revised manuscript). Nevertheless, the absorption is weaker at 785 nm than at 633 nm which explains the slower kinetic rate observed in case of both Ag and Au at 785 nm excitations.

2. It would be beneficial for the authors to compare the differences in plasmon-driven reactions between Au nanoparticles (NPs) and Ag NPs in terms of their local surface plasmon resonance (LSPR) properties, hot electron energy, or other relevant factors.

COMMENT: *A discussion of the observed molecular transformation in terms of their local surface plasmon resonance has been incorporated in the revised manuscript on page 14:*

“To be noted, Fig. 5D shows the comparison of three rate constant values calculated for Au aggregates under 785 nm excitation as no reaction could be observed at the lower laser powers. Additionally, the reaction rate calculated under 633 nm laser excitation for both Ag and Au (Fig. 5C) is higher than that observed under 785 nm excitation (Fig. 5D). This can be attributed to the usually stronger plasmon absorbance band (reflected from our previous micro absorbance study on 40 nm particle aggregates) at 633 nm than at 785 nm for Ag and Au aggregate structures relevant for our study.¹² The UV-vis spectra recorded for Ag dispersion coated with NMSB molecule after centrifugal wash also substantiate the fact, although the UV-vis spectral feature reflects the situation for NMSB-treated particles in solution. The UV-vis spectra for NMSB-treated AuNPs are also shown in Fig. S9B. Deconvolution of the UV-vis spectra (Fig.S9A) shows a secondary plasmon band centered at 640 nm (approx.) which creates a near resonance with a 633 nm laser excitation than a 785 nm source. This is indicative of the fact that the reaction rate depends on the availability of hot electrons (which is more favorable in the case of 633 nm than for 785 nm) and it was shown before that the hot electron generation rate is higher for Ag than Au, which can account for the higher reaction rates.⁴³”

3. The authors claim that the plasmon-driven reaction is initiated by hot-electrons rather than heat. Why external heat vs reaction rate can substantiate this claim? Why the influence of external heat is different for Au and Ag (Figure 5 E&F)?

COMMENT: The influence of temperature on the reaction and its rate was studied for two different reasons: The role of temperature in minimizing the activation barrier of reaction is well established. Therefore, the dominance of heat in the transformation process would have shown an increase in the reaction rate, which however is not observed. Secondly, a dominant effect of heat in the transformation of secondary product MBAm to MBN would lead to an enhanced dehydration pathway. Both Ag and Au don't show a clear dependence of external heat on reaction rate, therefore, we believe that the difference between Au and Ag is not significant.

4. Scheme 2 shows that the formation of p-mercaptobenzonitrile (MBN) is followed by the production of p-mercaptobenzamide (MBAm). This conclusion is drawn from Figure 2D (the SERS intensity time traces for $\nu(2230\text{ cm}^{-1})$ and $\nu(1193\text{ cm}^{-1})$), or from previous literature. Providing additional commentary on this point would enhance the clarity of the findings.

COMMENT: The formation of the product is evident from the SERS spectral changes observed during the transformation process. The formation of p-mercaptobenzonitrile (MBN) during the production of p-mercaptobenzamide (MBAm) is reported for the first time in this study.

5. The arrangement of Figures can be improved. For example, the information provided in Figures 1a and 1b are similar.

The magnification of SEM images for Ag and Au aggregates should be the same for comparison.

The label position of D1 and D2 in Figure 1 is not appropriate. It will be easier for readers if the author can label the reactant and products in D1 (NMSB) and D2 (e.g. MBN, MBAm).

The caption in Figure 2 only mentions the laser power, what's the laser spot area?

COMMENT: We thank the reviewer for the valuable suggestions. The above comments have been taken care of in the revised manuscript.

6. Carefully check the typos: For example, “binding motive”? In Figure 2, the label is 1189 cm⁻¹ in A and B, but the time traces in D become 1193 cm⁻¹. “Four different excitation laser sources” should be “Two different excitation laser sources” The unit of temperature is not the same in lines 346 and 349.

COMMENT: All necessary changes have been incorporated in the revised manuscript.

Reviewer #3 (Remarks to the Author):

The submitted work describes Plasmon plasmon-driven conversion of the secondary amine N-methyl-4-sulfanylbenzamide (NMSB).

The time dependence and kinetics of the conversion are studied by surface-enhanced Raman spectroscopy (SERS) and discussed in context to dissociative electron attachment (DEA) of the same molecule in the gas phase. The conversion and its kinetics are convincingly deduced from the time evolution of the observed Raman bands. The authors argue that the underlying reaction is mediated by electron transfer from the plasmonic nanostructures to the anchored NMSB molecules and argue that the underlying attachment process is the same in the gas phase, but the relaxation of the so-formed transient negative ion (TNI) differs mainly due to absence of the dominating S-H dissociation channel when NMSB is anchored by the sulfur bond to the respective.

While the electron-induced mediation of this reaction is apparent, the π^* σ^* coupling mechanism leading to the -CH₃ dissociation is speculative but plausible and a justified proposal.

Overall, the results are significant and of interest to the readers of Communications Chemistry, the data is at large of good quality and its interpretation is plausible. However, the structure of the paper and individual paragraphs need improvement before publication. I recommend publication of the submission after the authors have considered the comment here below.

In general, the rearrangement of the text is highlighted with "*" here below. I also recommend that the authors give the manuscript a good read and attempt to simplify the sentence structure and refrain from ambiguous and general statements where these may be

avoided. This also applies to the introduction though the points raised below are limited to the results and discussion.

Also, please check plural/singular and the use of articles.

COMMENT: We are grateful for the positive evaluation of our work and we have carefully checked the language of the manuscript.

-Figure 2, caption:

The readability of the caption could be improved by using simple enumeration followed by the description of the respective panels instead of a mixture with flowing text, i.e., A): Description. B): Description...

COMMENT: The above concerns has been incorporated wherever possible in the revised manuscript.

-Figure 2:

The 1189 cm⁻¹ NH₂ rocking is shown at 1193 cm⁻¹ in in panel D. This is a bit confusing and not addressed in the text, while this vibration is noted at 1189 cm⁻¹ in the caption and the text.

COMMENT: The above mistake has been corrected in the revised manuscript.

Line 135

“In parallel, the reaction is accompanied by the subsequent formation of p-mercaptobenzonitrile...”

This is a bit of a contradiction in terms, either the formation of p-mercapto benzonitrile is parallel to that of p-mercaptobenzamide or subsequent. From the time dependence of the rise of the 2230 cm⁻¹ peak, I do not see any indication that the CN formation is after the diminishing 1320 cm⁻¹ or rise in 1189 cm⁻¹.

Please clarify this point.

COMMENT: As per our observation and understanding, MBAm formed upon laser illumination undergoes further reaction to produce MBN and hence the term subsequent is used. This is due to the fact that the vibrational band corresponding to MBN is not observed at the initial time of the reaction (reflected in Figure 2A) but starts rising in few seconds after the

formation of MBAm. Therefore, for the ease of the reader, we stroke out the term “parallel” and the sentence has been rephrased as follows:

“The reaction is accompanied by the formation of p-mercaptobenzonitrile (MBN) indicated by the peak rise due to nitrile stretching vibration at 2230 cm⁻¹,³³ (Fig. 2C and D) reported for the first time in this study.”

Also, please consider showing the SER spectra with the calculated spectra in fig S3 (could then be shown as line spectra), for better comparison...the spectra of the intact NMSB in panels A and B and after exposure in the other panels. This would help the comparison.

Also, in panel A the hydrogen is missing from the sulfur in the formula. And, the DFT MO's would better be placed in a separate figure. They do not contribute to the comparison of the calculated and measured spectra but make the figure crowded.

Figure 3:

Similar to Fig S3, for ease of comparison it would be good to show an overlay with the measured SARS in the respective panels, i.e. the intact NMSB in panel A and after extensive exposure in the other panels. A comparison with the color-coded time dependence is not very clear.

COMMENT: For the ease of comparison, SERS spectrum of NMSB on respective Ag and Au aggregates after extensive exposure has been shown in Figures S3A and S4A and the respective calculated Raman spectra are shown in other panels from Figure S3&4(B-G) respectively.

In addition, SERS spectra of NMSB on Ag aggregates after extensive exposure is already incorporated in Figure 3D for comparison with the calculated Raman spectra shown in Figures 3A, B, and C.

Line 170...

“...blue shift observed from wavenumbers 315 cm⁻¹ to 345 cm⁻¹ to support the subsequent conversion of p-mercaptobenzamide to p-mercaptobenzonitrile during the reaction course.” Please clarify what is meant by subsequent conversion...is this a secondary DEA process...is that reflected in the kinetics?

COMMENT: Please refer to the response above for details. We state that MBN is formed as a final product from the primary product (or intermediate) MBAm. This is not a secondary DEA process. We have clarified the sentence that now reads:

“This is in line with the observation made in SERS measurements and we infer the temporal blue shift observed from wavenumbers 315 cm^{-1} to 345 cm^{-1} to support the subsequent conversion of the primary product MBAm to MBN during the reaction course.^{35,36}”

Line 201

“...of the NMSB molecule to prove the experimentally observed p-mercaptobenzamide formation...”

Propose to reformulate this sentence to read:

“...of the NMSB molecule to further substantiate the p-mercaptobenzamide formation...”

COMMENT: Done as suggested.

Lines 231-232:

“Concurrently, the water molecules trapped within the plasmonic substrate can act as a proton source albeit its energetically sluggish reaction kinetics to undergo the subsequent transformation to form p-mercaptobenzamide.”

This sentence needs to be rewritten and given some substantiated meaning. i) it is not clear what event this one is concurrent to, ii) the water molecules trapped within the plasmonic substrate have not been discussed...is this common knowledge, iii) what is “energetically sluggish reaction kinetics” and iv) it is not clear what the transformation to form p-mercaptobenzamide is after.

The following sentence should also be revisited.

“This explains the plausible reaction channels active in our case to induce the observed chemical transformations (although the detailed stepwise reactivity of the NMSB molecule is out of the scope of this study).

This is a very general statement and is withdrawn within the concluding parenthesis. Please rephrase to clarify what reaction channels the sentence above “explains” and substantiate or put forward the causality in a clear way.

COMMENT: We rephrased the discussion in the manuscript as below:

“Following the dissociation of the N–C α bond, the addition of a proton leads to the formation of MBAm. Plausibly, water molecules trapped within the plasmonic substrate act as a proton source albeit its energetically sluggish reaction kinetics (due to a high water oxidation potential). This explains the plausible reaction channels active in our case to induce the observed chemical transformations.”

264 dependency -> dependence

COMMENT: Done.

Line 238...:

“That the reaction is not purely light-induced (i.e. without involvement of plasmonic nanoparticles) is clear from our experimental observation where no reaction of the NMSB molecule deposited on Si-substrate (without nanoparticles) was observed at 633 nm and 785 nm laser excitation (Fig. 5A, B) indicating the role of plasmon excitation. Further, to understand the role of power and external temperature on the reactivity of NMSB molecule and to comment on the thermal or non-thermal nature of the reaction, we deduce the reaction rate constant for the dissociative transformation of NMSB molecule to p-mercaptobenzamide.”

*I recommend placing this before discussing the 633 nm data as it applies equally at both wavelengths.

* I believe the paper would benefit from rearranging so that all the plasmon-driven reactions, including the kinetics, are discussed before turning to the DEA data discussion. It is distracting and does not benefit the structure or coherence to split the plasmon-driven chemistry up like this.

COMMENT: We completely understand the point-of-view of the reviewer, however, the DEA data provides an important argument about the feasibility of an electron-induced pathway that we would like to present before we discuss other mechanistic detail. In order to provide a smoother transition we modified the text after the DEA discussion:

“Additional mechanistic details can be obtained from SERS data recorded und various different conditions. The reaction was also monitored under...”

Figure 5.

Please state clearly in the caption that panels A and B are from a Si substrate and if possible, signify that in the panels. Also please be consistent with the terminology. The intensity vs time spectra are termed “Time-lapse SERS trajectory”, “The time evolution of the peaks” and “time trajectories”...maybe more...

COMMENT: The requested changes have been incorporated in the revised manuscript.

Figure 5, caption

The readability of the caption could be improved by using simple enumeration followed by the description of the respective panels instead of a mixture with flowing text, i.e., A): Description. B): Description...

Also clarify what those mW are referring to, and if this is the laser power, please also state that in other places where it is of relevance.

COMMENT: The requested changes have been incorporated in the revised manuscript.

Line 274... :

“A schematic depicting the hot-electron transfer mediated dissociative decomposition of NMSB at the plasmonic interface of Ag is shown in Scheme 1. The HOMO-LUMO orbitals of the NMSB molecule when anchored on the surface of Au and Ag are shown in Fig. S8.”

Please extend the discussion on scheme 1 to bring it into context with the discussion of the DEA process, and please clarify the relationship to the MOs shown in Fig S8. In S8 there are no orbital lobes on NMSB except for LUMO +8 (Ag) and LUMO +7 (Au).

COMMENT: *We rephrased the discussion as below:*

“A schematic depicting the hot-electron attachment mediated dissociative decomposition of NMSB at the plasmonic interface of Ag is shown in Scheme 1. The LUMO +8 (Ag9) and LUMO +7 (Au9) (also refer to Fig. S9) represent the accessible ligand state molecular orbitals for electron attachment. The metal-molecule hybrid states representing either the metal or ligand state molecular orbitals when anchored on the surface of Au and Ag are shown in Fig. S9.”

Lines 289 -292

“Additionally, we observe that the p-mercaptobenzamide formed as the main product in this study further undergoes a reaction to generate p-mercaptobenzonitrile (although with a low yield, Figure 2A and D) plausibly through a dehydration reaction channel. (Scheme 2) This is an important observation as it opens up a secondary reaction route for nitrile formation on the plasmonic surface under visible light irradiation.”

This is partly the answer to few of my comments here above and needs to be placed in context to these claims when they are put forward...at which time they appear unclear and not substantiated. At least, though late, here a hypothesis is offered for the subsequent formation of the nitrile.

Line 303

“In this context, the effect of external heat on the reactivity and reaction rate of NMSB was studied by monitoring the reaction at seven different temperatures between 25 °C and 85 °C (Fig. 5E and F). A slow increment in the reaction rate with increasing temperature was noted in the case of both Au and Ag under 633 nm laser excitation. No profound effect of external heat on the dehydration reaction yield of p-mercaptobenzonitrile was noted. This further evidenced the less-dominant role of heat in the conversion of NMSB to p-mercaptobenzamide and p-mercaptobenzonitrile”.

Looking at Fig. 5E and F, there is an apparent rise in k above 50 Deg C in both cases and at the Au particles, k has increased more than three fold (about 2x on Ag). The error bars are however

significant and I believe that the only conclusion that may be drawn here is that no significant temperature effect may be deduced from this data.

***COMMENT:** We agree that the trend reflected from the k versus temperature plot and the significant error bar does not allow a clear interpretation of the temperature effect on the reaction rate. We rephrase the argument as below:*

“A slow increment in the reaction rate with increasing temperature was noted although the rise at 50 °C and 90 °C is quite apparent. However, the error bars in the case of both Au and Ag reflect a significant variation which does not allow us to conclude on the clear effect of temperature on the reaction rate.”

REVIEWERS' COMMENTS:

Reviewer #1 (Remarks to the Author):

The paper titled “Plasmon Driven Chemical Transformation of a Secondary Amide Probed by Surface Enhanced Raman Scattering” by Dutta et al. has undergone significant improvement since the previous submission. My comments were sufficiently addressed, and the detailed discussion brings a lot of important insights from the perspective of plasmon catalysis, making it relevant for the readers of Communications Chemistry. The description of the methodology used is now sufficient for a proper understanding of the experiment and its potential replication.

I would recommend incorporating the following author comments into the manuscript itself:

- a) “The reaction under study is assumed to be irreversible because the reaction products are likely to diffuse away or even desorb, and hence a returnable reaction is not favourable in this case.”
- b) “Considering fast excitation and relaxation of hot-charge carriers under continuous wave (CW) illumination, the time-average concentration of the hot electrons is considered constant which allows us to consider the process to follow a pseudo-first-order rate law.”

These comments clearly justify the approach used by the authors in this study, thus making the work more sound for the reader.

Additionally, I would recommend adding results obtained on the bare nanoparticles in the Supplementary Information (SI) to rule out the possibility of contamination for the reader.

Nevertheless, these minor issues do not change my opinion about the current form of the manuscript, which I consider significant for the field of plasmon catalysis. Therefore, I recommend its acceptance or acceptance with minor revisions for publication in Communications Chemistry.

Reviewer #2 (Remarks to the Author):

The authors have addressed all my comments, so I recommend the manuscript to be accepted for publication.

Reviewer #3 (Remarks to the Author):

I feel that the points raised in the previous round of review have at large been satisfactorily addressed and recommend to accept the manuscript for publication with few minor changes.

I recommend that the authors address the two minor points listed here below:

1. P11, L12-15: "Following a well-known mechanism in DEA,^{39,40} we can propose the capture of the electron into a π^* orbital of the aromatic moiety, followed by an electron transfer of the excess charge into the dissociative $\sigma^*(\text{S-H})$ orbital."

I find phrasing here a bit peculiar and propose to rather say:

"...followed by coupling with the antibonding $\sigma^*(\text{S-H})$ orbital.

2. P17, bottom: "A slow increment in the reaction rate with increasing temperature was noted although the rise at 50 °C and 90 °C is quite apparent."

In the figure the rise at 633 nm appears to be at 55 °C and for the 785 nm at 75 °C, or maybe 85 °C...90 °C is outside the measured range. Please also add the wavelength (785 nm) in panel F of Fig. 5.

Reviewer #1 (Remarks to the Author):

Comment:

The paper titled “Plasmon Driven Chemical Transformation of a Secondary Amide Probed by Surface Enhanced Raman Scattering” by Dutta et al. has undergone significant improvement since the previous submission. My comments were sufficiently addressed, and the detailed discussion brings a lot of important insights from the perspective of plasmon catalysis, making it relevant for the readers of Communications Chemistry. The description of the methodology used is now sufficient for a proper understanding of the experiment and its potential replication.

I would recommend incorporating the following author comments into the manuscript itself:

a) “The reaction under study is assumed to be irreversible because the reaction products are likely to diffuse away or even desorb, and hence a returnable reaction is not favourable in this case.”

b) “Considering fast excitation and relaxation of hot-charge carriers under continuous wave (CW) illumination, the time-average concentration of the hot electrons is considered constant which allows us to consider the process to follow a pseudo-first-order rate law.”

These comments clearly justify the approach used by the authors in this study, thus making the work more sound for the reader.

Response:

Both sentences have been included in the main manuscript now.

Comment:

Additionally, I would recommend adding results obtained on the bare nanoparticles in the Supplementary Information (SI) to rule out the possibility of contamination for the reader.

Nevertheless, these minor issues do not change my opinion about the current form of the manuscript, which I consider significant for the field of plasmon catalysis. Therefore, I recommend its acceptance or acceptance with minor revisions for publication in Communications Chemistry.

Response:

The spectra of bare nanoparticles have now been added as Figure S11 into the SI.

Reviewer #3 (Remarks to the Author):

Comment:

I feel that the points raised in the previous round of review have at large been satisfactorily addressed and recommend to accept the manuscript for publication with few minor changes.

I recommend that the authors address the two minor points listed here below:

1. P11, L12-15: "Following a well-known mechanism in DEA,^{39,40} we can propose the capture of the electron into a π^ orbital of the aromatic moiety, followed by an electron transfer of the excess charge into the dissociative $\sigma^*(S-H)$ orbital."*

I find phrasing here a bit peculiar and propose to rather say:

"...followed by coupling with the antibonding $\sigma^(S-H)$ orbital."*

Response:

The sentence has been changed as requested.

Comment:

2. P17, bottom: "A slow increment in the reaction rate with increasing temperature was noted although the rise at 50 °C and 90 °C is quite apparent."

In the figure the rise at 633 nm appears to be at 55 °C and for the 785 nm at 75 °C, or maybe 85 °C...90 °C is outside the measured range. Please also add the wavelength (785 nm) in panel F of Fig. 5.

Response:

Yes, the reviewer is correct, we have corrected the statement in the manuscript to 55 °C and 85°C. However, Figure 5F refers to 633 nm and AgNPs, which is explained in the figure caption and is indicated in the figure itself.

Furthermore, we are very grateful that the Reviewer has checked carefully our text again and we have incorporated most of the suggested changes.